# Enhancing Temporal Consistency in Video Editing by Reconstructing Videos with 3D Gaussian Splatting

**Inkyu Shin**[*]                                                    *dlsrbgg33@gmail.com*
*Korea Advanced Institute of Science and Technology*

**Qihang Yu**                                                    *qihang.yu@bytedance.com*
*ByteDance*

**Xiaohui Shen**                                                    *shenxiaohui@bytedance.com*
*ByteDance*

**In So Kweon**                                                    *iskweon77@kaist.ac.kr*
*Korea Advanced Institute of Science and Technology*

**Kuk-Jin Yoon**                                                    *kjyoon@kaist.ac.kr*
*Korea Advanced Institute of Science and Technology*

**Liang-Chieh Chen**                                                    *liangchieh.chen@bytedance.com*
*ByteDance*

**Reviewed on OpenReview:** *https://openreview.net/forum?id=s1zfBJysbI*

## Abstract

Recent advancements in zero-shot video diffusion models have shown promise for text-driven video editing, but challenges remain in achieving high temporal consistency. To address this, we introduce Video-3DGS, a 3D Gaussian Splatting (3DGS)-based video refiner designed to enhance temporal consistency in zero-shot video editors. Our approach utilizes a two-stage 3D Gaussian optimizing process tailored for editing dynamic monocular videos. In the first stage, Video-3DGS employs an improved version of COLMAP, referred to as MC-COLMAP, which processes original videos using a Masked and Clipped approach. For each video clip, MC-COLMAP generates the point clouds for dynamic foreground objects and complex backgrounds. These point clouds are utilized to initialize two sets of 3D Gaussians (Frg-3DGS and Bkg-3DGS) aiming to represent foreground and background views. Both foreground and background views are then merged with a 2D learnable parameter map to reconstruct full views. In the second stage, we leverage the reconstruction ability developed in the first stage to impose the temporal constraints on the video diffusion model. This approach ensures the temporal consistency in the edited videos while maintaining high fidelity to the editing text prompt. We further propose a recursive and ensembled refinement by revisiting the denoising step and guidance scale used in video diffusion process with Video-3DGS. To demonstrate the efficacy of Video-3DGS on both stages, we conduct extensive experiments across two related tasks: Video Reconstruction and Video Editing. Video-3DGS trained with 3k iterations significantly improves video reconstruction quality (+3 PSNR, +7 PSNR increase) and training efficiency (×1.9, ×4.5 times faster) over NeRF-based and 3DGS-based state-of-art methods on DAVIS dataset, respectively. Moreover, it enhances video editing by ensuring temporal consistency across 58 dynamic monocular videos. Project website is available here.

---

[*]Work done during internship at ByteDance.

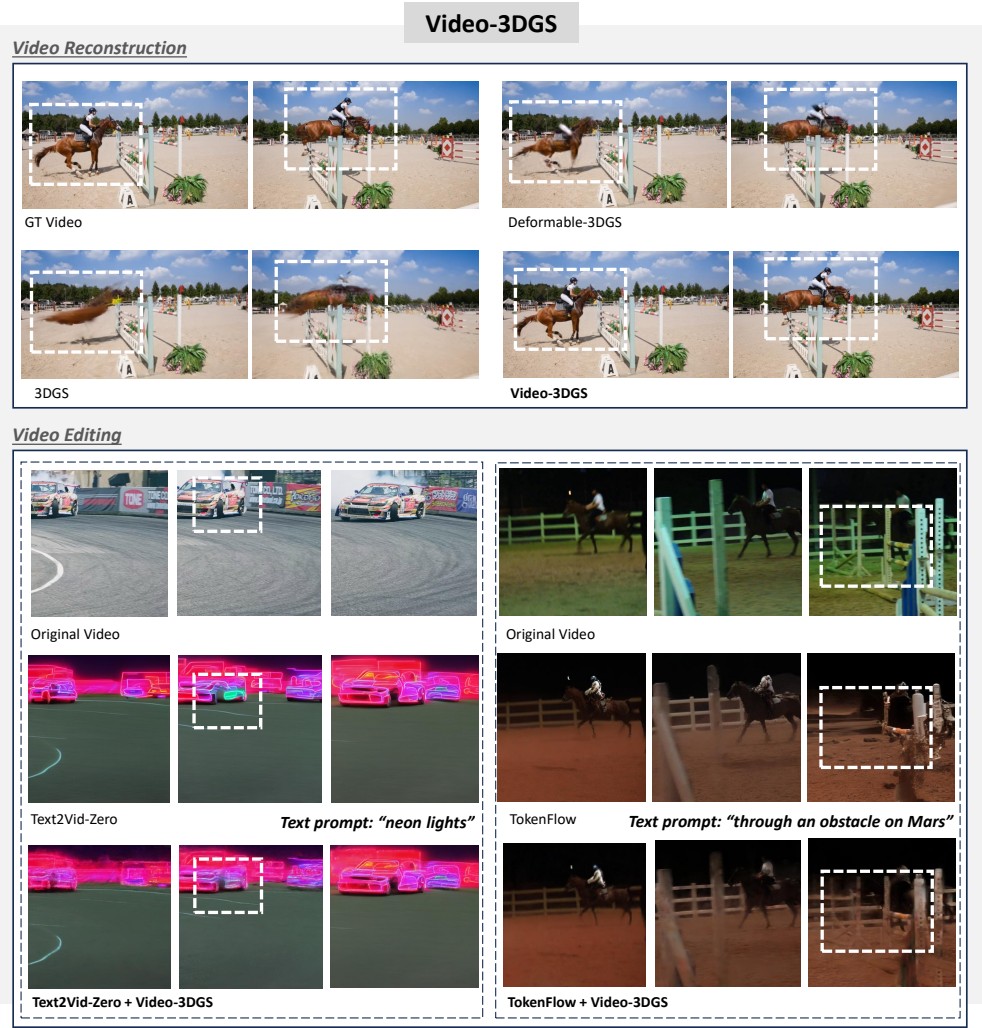

Figure 1: The proposed Video-3DGS expands the capabilities of 3D Gaussian Splatting (3DGS) (Kerbl et al., 2023) to dynamic monocular video scenes, enhancing temporal consistency in both video reconstruction and video editing. For instance, it consistently captures and reconstructs dynamic objects such as riders and horses (upper section), while also enriching style smoothness in scenarios like drift-car sequences (bottom left) and ensuring structure consistency (bottom right). Regions of interest are highlighted by white dashed rectangles.

# 1    Introduction

Advancements in diffusion-based generative models (Ho et al., 2020; Rombach et al., 2022) have significantly improved text-driven image editing capabilities (Brooks et al., 2023). Building on this success, there has been considerable effort to extend these technologies to the video domain (Ho et al., 2022; Singer et al., 2022), which holds practical potential across a broad range of applications, including Film/Entertainment and AR/VR. However, training a video diffusion model from scratch using video datasets is not effective for two main reasons: (i) it lacks a well-constructed video training dataset to handle the diverse distribution of videos in the wild, and (ii) discarding pre-trained image diffusion models would incur a penalty in generating high-quality edited frames for videos. To address the limitations, zero-shot training-free video editing methods (Khachatryan et al., 2023; Geyer et al., 2023; Kara et al., 2023) that are built on pre-trained image diffusion model have recently been introduced to not only increase the video editing quality but also improve

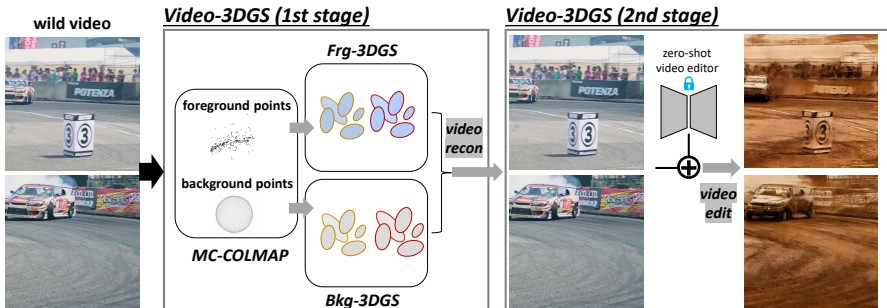

Figure 2: The overall pipeline of Video-3DGS. We aim to design a video-level 3D Gaussian Splatting framework to reconstruct the video scenes (1st stage), which enables high temporal consistency in video editing (2nd stage). Specifically, Video-3DGS is empowered by the proposed MC-COLMAP that effectively obtains 3D points for foreground moving objects. The background 3D points are modeled with spherical-shaped random points, surrounding the foreground points. Video-3DGS utilizes two sets of 3D Gaussians (Frg-3DGS and Bkg-3DGS) to represent foreground and background 3D points, respectively. A 2D learnable parameter map merges the foreground and background views, rendered from each set of 3D Gaussians. The merged views enable high-fidelity video reconstruction. Then, we leverage this reconstruction capability into zero-shot video editor to enhance temporal consistency while maintaining high fidelity to text prompt.

efficiency. Nevertheless, these zero-shot video diffusion models still exhibit low temporal consistency in dynamic videos due to their limited understanding of individual video scenes. A natural question thus emerges: *Is it possible to design a simple yet effective plug-and-play module to enhance the temporal consistency of each edited video from any zero-shot video editors?*

To answer the question, we carefully design Video-3DGS, an innovative approach that leverages per-scene representation power of 3D Gaussian Splatting (3DGS) (Kerbl et al., 2023) to enhance temporal consistency ability for zero-shot video diffusion models. Our approach aims to utilize the structural preservation ability of 3DGS, which is known for explicitly representing and associating multiple views with unified 3D Gaussians. As illustrated in Figure 2, Video-3DGS employs a two-stage 3DGS optimization process to reconstruct dynamic monocular videos and refine the initially edited ones, ensuring enhanced temporal consistency and visual coherence.

The first stage begins by addressing the limitations of 3DGS in representing dynamic monocular video scenes, where multiple objects move against complex backgrounds. To solve this issue, new methods adopting 3DGS for dynamic monocular videos are emerging. Deformable-3DGS (Yang et al., 2023) employed the deformation network to condition 3DGS with time-variable, while 4DGS (Wu et al., 2023a) additionally added HexPlane (Cao & Johnson, 2023) encoder before the deformation network. Despite these attempts to extend 3DGS for monocular video scenes, it is important to acknowledge the persisting limitations, which are summarized as follows. (i) The fragile dependency on the underlying SfM method, *e.g.*, COLMAP (Schönberger & Frahm, 2016) for deriving 3D points from the entire dynamic videos. (ii) The difficulty of accurately representing monocular videos with a single set of 3D Gaussians. Video frames, particularly those with dynamic motion and intricate background, are infeasible to be represented by a single set of 3D Gaussians, even with the deformation network. As we can observe from Figure 1, Deformable-3DGS (Yang et al., 2023) shows unsatisfying video reconstruction results for dynamic moving objects. Therefore, we first design the framework of Video-3DGS to address the complexities of dynamic monocular video scenes. One of the key distinctions of the proposed method lies in its ability to excel in "wilder" monocular video scenarios (*e.g.*, videos with larger object motion). To this end, we initially simplify the video sequences with two decomposition strategies. First, spatial decomposition, powered by an off-the-shelf open vocabulary video segmentor, mitigates background clutter [1]. Second, temporal decomposition breaks down the entire video sequence into multiple shorter video clips with overlapping frames between neighboring clips. Given these

---

[1]background containing complex regions that clutter the foreground objects (*e.g.*, trees in the background).

decompositions, we can effectively extract 3D points of masked foreground moving objects in each video clip by introducing MC-COLMAP. Meanwhile, the cluttered background is modeled with spherical-shaped random 3D points surrounding the pre-extracted 3D points of foreground moving objects. For each video clip, Video-3DGS utilizes two sets of 3D Gaussians (Frg-3DGS and Bkg-3DGS) to represent foreground and background 3D points, respectively. We additionally implement multi-resolution hash encoding (Müller, 2021) with deformation networks (Yang et al., 2023) on both Frg-3DGS and Bkg-3DGS, which can boost performance and efficiency. Last but not least, to obtain the final rendered 2D outputs from both 3D Gaussians, we adopt a straightforward and effective 2D learnable parameter map to merge the two rendered images (rendered views from Frg-3DGS and Bkg-3DGS), resulting in faithful video frame representations.

The subsequent stage focuses on seamlessly integrating a pre-optimized Video-3DGS into existing zero-shot video editors. The primary advantage of using a pre-optimized Video-3DGS lies in its ability to apply structured constraints of 3D Gaussians across multiple video frames. More specifically, we maintain the structural components of Frg-3DGS and Bkg-3DGS in a fixed state while selectively fine-tuning the color parameters, such as spherical harmonic coefficients, alongside a 2D learnable parameter map. This fine-tuning process is designed to capture and replicate the style of the initially edited video frames. By applying fixed 3D Gaussian structures with style updates in each clip and smoothing transitions between overlapping frames across neighboring clips, we effectively minimize style flickering throughout the entire video, enhancing temporal consistency. We have also observed that existing zero-shot video editors exhibit sensitivity to variations in parameters, such as the number of denoising steps and the scale of image or text guidance. These variations can significantly impact the editing outputs, as demonstrated in Figure 3. To optimize effectiveness and reduce parameter sensitivity, we introduce a recursive and ensembled video editing strategy. This involves interspersing Video-3DGS between split denoising steps and updating styles using multiple videos edited under different guidance scales. This further exploration helps to stabilize the editing outcomes across varying parameter settings.

To validate the proposed two-stage Video-3DGS optimization, we tested on corresponding video tasks: video reconstruction and video editing. For video reconstruction, our approach surpasses both NeRF-based and 3DGS-based state-of-the-art methods across 28 DAVIS videos. We then demonstrate the applicability of Video-3DGS's video scene representation to the video editing task in 58 challenging monocular videos sourced from the CVPR 2023 LOVEU Text-Guided Video Editing (TGVE) challenge (Wu et al., 2023b). Video-3DGS consistently enhances editing quality across three off-the-shelf video editors (Text2Video-Zero(Khachatryan et al., 2023), TokenFlow (Geyer et al., 2023), and RAVE (Kara et al., 2023)).

## 2 Method

The meta-architecture of Video-3DGS aims to design a video-specific 3DGS, which serves as a plug-and-play refiner for initially edited video from zero-shot video editors. This process is structured into two seamless stages. The first stage of Video-3DGS (Section 2.2) aims to represent and reconstruct original videos with two integrated components: MC-COLMAP (Section 2.2.1) and Foreground/Background 3DGS (Section 2.2.2). MC-COLMAP plays a pivotal role in generating masked clip-level foreground and background 3D points. Subsequently, for each clip, two sets of 3D Gaussians, Frg-3DGS and Bkg-3DGS, are initialized and optimized based on these points. Additionally, a 2D learnable parameter map is employed to merge the foreground and background views rendered from Frg-3DGS and Bkg-3DGS. The resulting views accurately represent the video frames, facilitating the video reconstruction. Section 2.3 details the second stage, where we transform the optimized Video-3DGS into a plug-and-play temporal refiner for video editing.

### 2.1 Preliminary

**Diffusion Model** Diffusion probabilistic models (DPM), introduced by (Sohl-Dickstein et al., 2015) and further developed in (Ho et al., 2020) represent a class of generative models designed to approximate a data distribution $q$ through a progressive denoising process. The model starts with a Gaussian i.i.d noisy image $x_T \sim \mathcal{N}(0, I)$ and the diffusion model $\epsilon_\theta$ incrementally denoises this image until it reaches a clean image $x_0$ drawn from the target distribution $q$. The Deterministic Sampling algorithm (DDIM) described in (Song et al., 2020) initializes the noise process, termed DDIM inversion, by starting from a clean image $x_0$. This

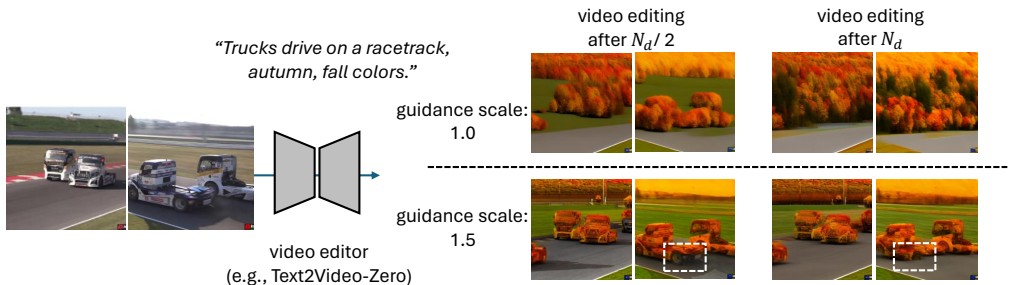

Figure 3: We revisit the key hyperparameters in the video diffusion process: the denoising step ($N_d$) and the guidance scale. Employing a higher denoising step combined with a lower guidance scale (*e.g.*, similar to the image guidance scale in Text2Video-Zero (Khachatryan et al., 2023)) results in greater fidelity to the editing prompt but compromises structural and temporal consistency, and vice versa. This analysis confirms that the zero-shot video editor model is highly sensitive to these hyperparameters.

approach effectively facilitates image editing through diffusion models. Subsequently, the technique has been adapted for video editing by incorporating temporal constraints. This extension employs several strategies to ensure consistency across frames, including the use of a 3D UNet with cross-frame attention as explored in Text2Video-Zero (Khachatryan et al., 2023), diffusion feature propagation as discussed in TokenFlow (Geyer et al., 2023), and the grid trick detailed in RAVE (Kara et al., 2023).

**3D Gaussian Splatting**  We also present an overview of 3D Gaussian Splatting (3DGS) (Kerbl et al., 2023). The method involves the initialization of a set of 3D Gaussians upon 3D point clouds. The point clouds are derived from Structure from Motion (SfM) techniques, exemplified by COLMAP (Schönberger & Frahm, 2016) with the input $\mathcal{V}$ of a series of $N$ consecutive frames $\mathbf{x}_i$ (*i.e.*, $\mathcal{V} = \{\mathbf{x}_i\}_{i=1}^{N}$) in a monocular video sequence. Then, 3D Gaussians are trained with the function of $G(x, r, s, \sigma, SH)$ containing center position $x$, 3D covariance matrix obtained from quaternion $r$ and scaling $s$, opacity $\sigma$, and spherical harmonic coefficient $SH$.

## 2.2 Video-3DGS (1st Stage): Reconstructing Videos with 3DGS

### 2.2.1 MC-COLMAP

We denote the conventional COLMAP function as $R$, which is designed to derive a 3D point cloud $p$, along with the associated camera data $c$ (encompassing both intrinsic ($c^{in}$) and extrinsic ($c^{ex}$) parameters). Furthermore, it outputs a feedback signal *status*, which serves as an indicator of the COLMAP system's success in reconstructing 3D points from the entire video frames $\mathcal{V}$. That is,

$$p, c, status = R(\mathcal{V}) \tag{1}$$

However, the presence of foreground moving objects and cluttered background in video frames poses a significant challenge in extracting consistently matched 3D points across the entire video frames. This difficulty leads to inaccurate and sparse reconstructions in video outputs, as is evident in our experiments. To tackle these challenges, we propose MC-COLMAP, a revised version of COLMAP, specifically designed to progressively process video frames. It effectively minimizes motion and complex background through two key strategies: spatial decomposition and temporal decomposition.

**Spatial Decomposition**  To reduce the background cluttering effect, we adopt an off-the-shelf open-vocabulary video object segmentation network $S$ (*e.g.*, DEVA (Cheng et al., 2023)) to extract the segmentation masks for foreground moving objects in the video $\mathcal{V}$ as follows:

$$\mathcal{V}^f = S(\mathcal{V}, class) \tag{2}$$

where $\mathcal{V}^f$ and *class* denote the extracted segmentation masks of foreground moving objects and the user-guided text prompt (a required input for the segmentation network to specify the target object), respectively.

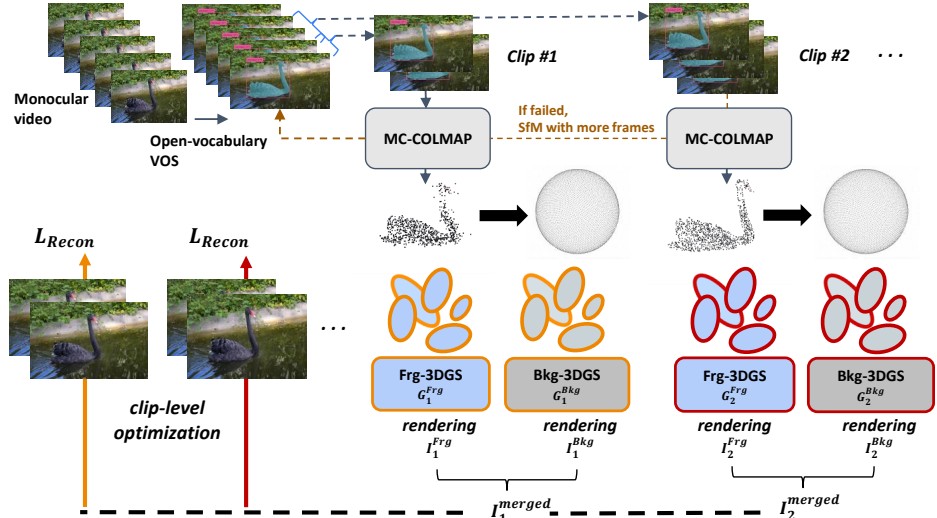

Figure 4: The proposed Video-3DGS (1st stage) comprises two key components. First, to effectively capture 3D point clouds and corresponding frame viewpoints from dynamic monocular videos, we introduce **M**asked and **C**lipped COLMAP (MC-COLMAP). This module spatially and temporally decomposes video frames, facilitating the extraction of clip-level foreground points through progressively processing clips. Additionally, we initialize spherical-shaped random background points conditioned on the foreground points. Second, with these two sets of point clouds, we introduce two distinct sets of 3D Gaussian Splatting (3DGS): Frg-3DGS and Bkg-3DGS, optimized separately for foreground and background points, respectively. Subsequently, we employ a straightforward merging operation to combine the rendered outputs of Frg-3DGS and Bkg-3DGS. We optimize the merged rendered outputs for each clip using the reconstruction loss.

For example, in the top left of Figure 4, we can consistently extract 'Swan' across frames in monocular video using spatial decomposition strategy.

**Temporal Decomposition** The segmentation masks $\mathcal{V}^f$ for foreground objects, which span the entire video sequence, present a processing challenge for COLMAP due to their intricate motion dynamics. To mitigate this complexity, stemming from diverse motion patterns throughout the video sequence, we partition the video sequence into multiple shorter video clips. This division ensures that objects exhibit reduced motion within each clip, thus facilitating more manageable processing for COLMAP. Formally, we address this by splitting $\mathcal{V}^f$ into multiple $M$ clips $\{\mathcal{V}_j^f\}_{j=1}^M$.

Rather than evenly dividing the video sequence into $M$ clips, each containing $k$ frames, we introduce a progressive scheme to address potential failure cases, such as when the foreground object remains static within a clip, making the $k$ frames inadequate for point cloud extraction or registration in SfM. Specifically, we begin with the first clip initialized with the first $k$ frames (*i.e.*, $\mathcal{V}_1^f = \{\mathbf{x}_i\}_{i=1}^k$). If the COLMAP function $R$ fails to process the clip (*i.e.*, $status_1 \neq$ 'Success'), we iteratively include one additional consecutive frame into the current clip until COLMAP returns a 'Success' *status*. Subsequently, the next clip commences with the last frame of previous clip, also initialized with $k$ frames. This process continues until the entire video sequence is processed. We term the resulting pipeline as MC-COLMAP, which applies COLMAP in a **M**asked and **C**lipped manner. Our MC-COLMAP system (denoted as $R_{MC}$) yields multiple sets of masked and clipped 3D point clouds, along with their corresponding views from $M$ clips as demonstrated in the top right of Figure 4 and following equation:

$$\{(p_j, c_j)\}_{j=1}^M = R_{MC}(S(\mathcal{V}, class)) \tag{3}$$

We validate the effectiveness of MC-COLMAP in Table 1 and Table 3, demonstrating that MC-COLMAP not only improves the success rate over the original COLMAP, but also provides better-initialized 3D points for constructing 3D Gaussians.

### 2.2.2 Foreground and Background 3DGS

**Foreground 3D Gaussians** For each video clip, the 3D point clouds for foreground moving objects, derived from Equation (3), serve as initialization for optimizing foreground 3D Gassuians. This process yields a set of 3D Gaussians $\{G_j^{Frg}\}_{j=1}^M$, tailored for those foreground point clouds with the corresponding camera views, $\{(p_j, c_j)\}_{j=1}^M$. With this, we successfully represent the foreground objects using 3D Gaussians, and the subsequent step involves modeling the cluttered background.

**Background 3D Gaussians** For each video clip, to simply model the cluttered background, we utilize spherical-shaped random point clouds $\{p_j^{Bkg}\}_{j=1}^M$, surrounding the previously extracted foreground 3D points. These background random point clouds serve as the initialization for optimizing background 3D Gaussians, yielding a set of 3D Gaussians $\{G_j^{Bkg}\}_{j=1}^M$. Notably, the spherical-shaped random point clouds are defined by two hyper-parameters: number of points $n_{Bkg}$ and its radius $r_i$. Our method is empirically found robust to those hyper-parameters, and we fix $n_{Bkg} = 60k$ points and $r_i$ to 3 times larger than the foreground points' distance that is measured by the maximum Euclidean distance between foreground points.

**Deformable 3D Gaussians** Following (Yang et al., 2023), both the foreground and background 3D Gaussians are enhanced with the deformable network, which effectively leverage the time information, but we extend them for clip-level processing, resulting in:

$$\begin{aligned} \text{Frg-3DGS: } &\{G_j^{Frg}(\{x_j, r_j, s_j\} + \delta_j^{Frg}, \sigma_j, SH_j)\}_{j=1}^M \\ \text{Bkg-3DGS: } &\{G_j^{Bkg}(\{x_j, r_j, s_j\} + \delta_j^{Bkg}, \sigma_j, SH_j)\}_{j=1}^M \end{aligned} \tag{4}$$

where $\delta_j$ is the deformation within the $j$-th clip to transform center $x_j$, rotation $r_j$, and scale $s_j$, according to each clip center and normalized time. The superscript $Frg$ and $Bkg$ denote foreground and background, respectively. We implement the deformation network with 4D multi-resolution hash encoding (Müller, 2021). Given those two sets of Frg-3DGS and Bkg-3DGS, we reformulate them as Clip-3DGS, where each clip contains its corresponding Frg-3DGS and Bkg-3DGS. That is, Clip-3DGS = $\{G_j^{Frg}, G_j^{Bkg}\}_{j=1}^M$. We then process those 3D Gaussians in a clip-by-clip manner.

**Merging Foreground and Background Views with 2D Learnable Parameters** For the $j$-th clip, we leverage the differentiable point-based rendering technique (Wiles et al., 2020), as outlined in (Kerbl et al., 2023), to process the two sets of 3D Gaussians $G_j^{Frg}$ and $G_j^{Bkg}$. This enables us to generate two distinct rendered images for each frame within the $j$-th clip. Specifically, rendering the $i$-th video frame produces two images, $\{I_i^{Frg}, I_i^{Bkg}\}$, derived from $G_j^{Frg}$ and $G_j^{Bkg}$, respectively. To seamlessly merge these images, both of dimensions height $H$ and width $W$, we introduce a straightforward yet powerful merging technique. This method employs a 2D learnable parameter $\alpha \in \mathcal{R}^{H \times W}$, facilitating pixel-wise merging with corresponding learnable parameters initialized to a value of 0.5. Formally, we have:

$$I_i^{merged} = \alpha_i \times I_i^{Frg} + (1 - \alpha_i) \times I_i^{Bkg} \tag{5}$$

where $I_i^{merged}$ is the merged result for the $i$-th video frame.

Through the merging operation, we derive merged images for all video frames by optimizing $N$ different $\alpha$ values ($N$ represents the total number of frames in the video).

**Training Losses** To optimize Frg-3DGS and Bkg-3DGS, we adopt the reconstruction loss, denoted as $L_{recon}$, which comprises two components: $L_1$ and $L_{SSIM}$, akin to the approach outlined in (Kerbl et al., 2023). These components are calculated by comparing three rendered images - the foreground $I^{Frg}$, the background $I^{Bkg}$, and the merged image $I^{merged}$ - against their respective ground truth images.

**Video Reconstruction** The reconstruction of a video can be achieved by sequentially rendering the Frg-3DGS and Bkg-3DGS for each clip. Subsequently, the rendered images from each frame are merged using a pre-trained alpha parameter (Equation (5)).

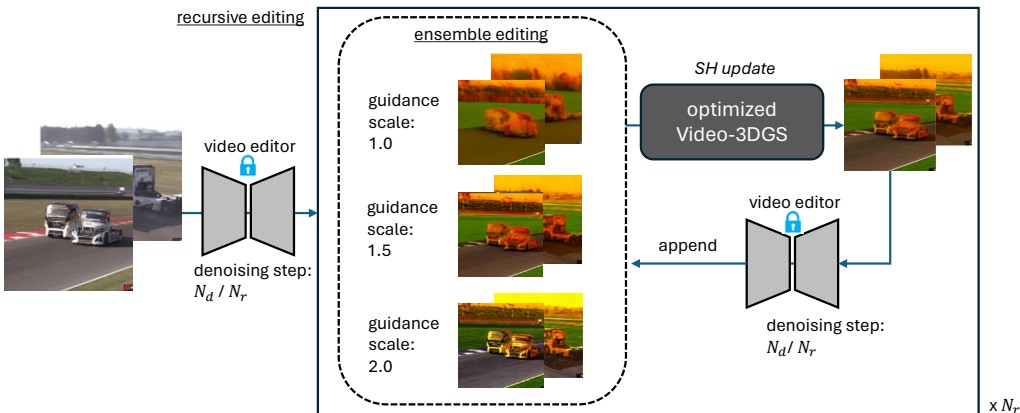

Figure 5: The overview of Video-3DGS (2nd stage) as a plug-and-play refiner for video editing begins with fine-tuning the spherical coefficient of the optimized Video-3DGS on an initially edited video, which is produced using an off-the-shelf video editor with the default hyperparameters: a denoising step $N_d$ and a guidance scale. This method, referred to as the single-phase refiner with Video-3DGS, is further enhanced by our findings in Figure 3. We split $N_d$ into a recursive number $N_r$ and fine-tune the spherical coefficient parameters against multiple outputs from varied guidance scales, aiming for improved temporal consistency and high fidelity to the editing text. This advanced approach is named the Recursive and Ensembled (RE) refinement.

## 2.3 Video-3DGS (2nd Stage): Plug-and-Play Refiner for Editing Videos

**Single-phase Refiner** We employ pre-trained 3D Gaussians to preserve the original structure and ensure temporal consistency in video editing. Edited video frames, denoted as

$$\{I_i^{edited}\}_{i=1}^N, \tag{6}$$

can be generated by any zero-shot video editor. However, these edits may introduce style inconsistencies or cause original objects to be changed. To address this issue, we fix the positional parameters $(x, r, s)$ and deformation parameters $(\delta)$ of both Frg-3DGS and Bkg-3DGS. At the same time, we adjust the color value $(SH)$ and opacity $(\sigma)$ parameters to match the edited style. This update is achieved by minimizing the reconstruction loss between the rendered images $\{I_i^{merged}\}_{i=1}^N$ and edited frames $\{I_i^{edited}\}_{i=1}^N$ for updating corresponding $SH$ and $\sigma$ as follows:

$$\min_{\mathbf{SH}, \sigma} \sum_{i=1}^N L_{\text{recon}}(I_i^{merged}, I_i^{edited}) \tag{7}$$

Adjusting the color values ensures that corresponding regions in the rendered images remain consistent within each clip. Furthermore, by refining overlapping frames between adjacent clips, we achieve smoother style transitions and improved temporal consistency throughout the video. Given that we adopt Video-3DGS following a single-phase of video editing, Video-3DGS (2nd stage) can be appropriately referred to as a single-phase refiner.

**Recursive and Ensembled Refiner** In the single-phase refiner, we update $SH$ and $\sigma$ by minimizing the reconstruction loss between the rendered images and the edited frames. However, when the off-the-shelf video editor uses an excessive number of denoising steps $(N_d)$ with editing prompts, some edited frames may lose essential structural details. To counteract this, we split $N_d$ into multiple phases (with $N_r = 2$ for efficiency) and provide the detailed explanation for each phase.

*First Phase:* we first apply the signle-phase refinement Equation (7) on the initially edited frames (e.g., after $N_d/2$ denoising steps). This produces a set of intermediate refined frames.

| Method | Metrics | success rate | PSNR | SSIM | Time |
|---|---|---|---|---|---|
| NeRF-based | Robust-Dyn (Liu et al., 2023) | 28 / 28 | 26.8 | 0.795 | 746m |
| | NLA (Kasten et al., 2021) | 28 / 28 | 27.3 | 0.795 | 309m |
| | CoDeF (Ouyang et al., 2023) | 28 / 28 | 29.4 | 0.869 | 28m |
| | Nerv (Chen et al., 2021b) | 28 / 28 | 34.6 | 0.980 | 20m |
| 3DGS-based | 3DGS (Kerbl et al., 2023) | 20 / 28 | 24.8 | 0.832 | 10m |
| | Deform-3DGS (Yang et al., 2023) | 20 / 28 | 30.6 | 0.919 | 50m |
| 3DGS-based (ours) | Video-3DGS (3k) | 28 / 28 | 37.6 | 0.980 | 11m |
| | Video-3DGS (5k) | 28 / 28 | 41.2 | 0.989 | 22m |
| | Video-3DGS (10k) | 28 / 28 | 45.8 | 0.995 | 56m |

Table 1: Average quantitative results of DAVIS videos for Video Reconstruction. 'success rate' indicates the proportion of videos, out of the total 28 videos, that can be successfully reconstructed. Our method, Video-3DGS, demonstrates efficient training and significantly outperforms NeRF-based and 3DGS-based methods. We report Video-3DGS results trained with 3k, 5k, and 10k iterations.

*Initialization for the Next Phase:* we use the intermediate edited frames from above first phase as input for DDIM inversion (as in TokenFlow (Geyer et al., 2023) and RAVE (Kara et al., 2023) or an image encoder (as in Text2Video-Zero (Khachatryan et al., 2023)) to initialize the subsequent phase.

*Second Phase:* Then, we complete the remaining $N_d/2$ denoising steps for further video editing and follow with a final single-phase refinement on the newly edited frames.

We refer to this overall procedure as **recursive refinement** with multiple phases. Additionally, the guidance parameters in the diffusion process (e.g., image guidance in Text2Video-Zero, text guidance in TokenFlow, and ControlNet guidance in RAVE) play a crucial role in the quality of the edits. As shown in Figure 3, different guidance scales can lead to significant variations. To reduce sensitivity to these scale changes and combine the benefits of multiple scales, we perform an ensemble update of $SH$ and $\sigma$ using edited videos generated at three different scales during each recursive phase. The ensemble of edits from one phase is then incorporated into the next, allowing each phase to benefit from previous refinements. For simplicity, we denote the 2nd-stage Video-3DGS with **Recursive and Ensembled** refinement as Video-3DGS (RE), as illustrated in Figure 5.

## 3 Experimental Results

In this section, we evaluate Video-3DGS on two tasks: Video Reconstruction (Section 3.1) and Video Editing (Section 3.2). We provide the details of datasets and metrics used for those two tasks (Appendix B), ablation studies (Appendix D), and qualitative results (Appendix E) in the Appendix.

### 3.1 Video Reconstruction

**Baselines**  As Video-3DGS aims to reconstruct videos by learning scene representations, we compare our approach with two representative methods: NeRF-based and 3DGS-based representations. For NeRF-based methods, we select four state-of-the-art baselines: 1) NLA (Kasten et al., 2021), which proposes a video reconstruction and editing method using layered atlases powered by the NeRF framework. 2) CoDEF (Ouyang et al., 2023), which leverages a canonical space of deformation fields to reconstruct and edit videos. 3) RobustDyn (Liu et al., 2023), an advanced view reconstruction and synthesis model that estimates camera poses in diverse settings. 4) Nerv Chen et al. (2021b), which proposes neural representation to encode videos in neural networks. For 3DGS-based methods, we first consider the original 3DGS method (Kerbl et al., 2023), which lacks modules specific to video scenes. Additionally, we select Deformable-3DGS (Yang et al., 2023), a state-of-the-art approach that utilizes a deformation network on 3D Gaussian, serving as another strong baseline.

**Quantitative Results**  As illustrated in Table 1, our comprehensive video reconstruction experiments encompass 28 videos sourced from DAVIS. We note that the four NeRF-based methods generally exhibit limitations in both reconstruction quality (measured by PSNR and SSIM) and efficiency (training time), primarily attributed to their implicit neural representation. Conversely, the 3DGS method (Kerbl et al., 2023) demonstrates a significant reduction in training time (less than 10 minutes in average of 20 videos),

| Dataset \ Method | Text2Vid-Zero | | +Video-3DGS | | +Video-3DGS (RE) | |
|---|---|---|---|---|---|---|
| | WarpSSIM↑ | $Q_{edit}$↑ | WarpSSIM↑ | $Q_{edit}$↑ | WarpSSIM↑ | $Q_{edit}$↑ |
| DAVIS | 0.691 | 20.1 | 0.827 (+13.6%) | 21.0 (+0.9%) | 0.899 (+20.8%) | 22.3 (+2.2%) |
| Videovo | 0.773 | 21.9 | 0.902 (+12.9%) | 22.1 (+0.2%) | 0.926 (+15.3%) | 23.1 (+1.2%) |
| Youtube | 0.701 | 19.8 | 0.885 (+18.4%) | 20.4 (+0.6%) | 0.922 (+22.1%) | 21.1 (+1.3%) |

| Dataset \ Method | TokenFlow | | +Video-3DGS | | +Video-3DGS (RE) | |
|---|---|---|---|---|---|---|
| | WarpSSIM↑ | $Q_{edit}$↑ | WarpSSIM↑ | $Q_{edit}$↑ | WarpSSIM↑ | $Q_{edit}$↑ |
| DAVIS | 0.855 | 22.9 | 0.909 (+5.4%) | 23.9 (+1.0%) | 0.912 (+5.7%) | 24.8 (+1.9%) |
| Videovo | 0.897 | 22.6 | 0.933 (+4.6%) | 23.5 (+0.9%) | 0.937 (+4.9%) | 23.8 (+1.2%) |
| Youtube | 0.848 | 22.1 | 0.923 (+7.5%) | 23.1 (+1.0%) | 0.923 (+7.5%) | 24.2 (+2.1%) |

| Dataset \ Method | RAVE | | +Video-3DGS | | +Video-3DGS (RE) | |
|---|---|---|---|---|---|---|
| | WarpSSIM↑ | $Q_{edit}$↑ | WarpSSIM↑ | $Q_{edit}$↑ | WarpSSIM↑ | $Q_{edit}$↑ |
| DAVIS | 0.872 | 23.4 | 0.908 (+3.6%) | 24.1 (+0.7%) | 0.913 (+4.1%) | 24.8 (+1.5%) |
| Videovo | 0.872 | 22.4 | 0.913 (+4.1%) | 23.5 (+1.1%) | 0.923 (+5.1%) | 23.6 (+1.2%) |
| Youtube | 0.855 | 21.7 | 0.918 (+6.3%) | 22.9 (+1.2%) | 0.921 (+6.6%) | 23.5 (+1.8%) |

Table 2: Quantitative results of Video Editing on three different datasets: DAVIS, Videvo, Youtube. Each dataset contains four editing categories: style change, object change, background change, multiple change. We present the results for average WarpSSIM and $Q_{edit}$ of four categories, with text color indicating the effect of Video-3DGS. Our Video-3DGS increases both temporal consistency and overall video editing performance across all initial video editors. ↑: the higher, the better.

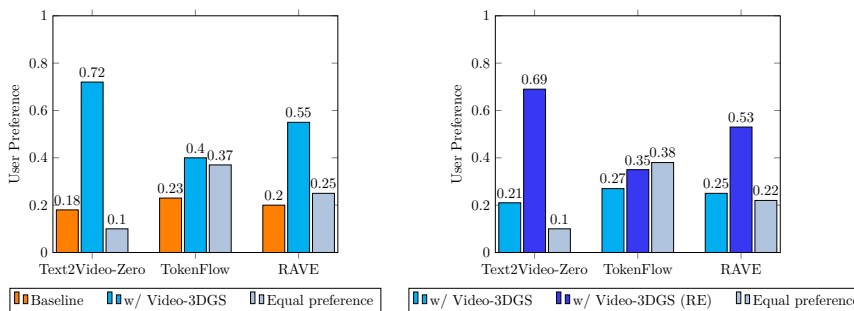

Figure 6: User study of Video-3DGS on three video editors.

owing to its explicit 3D Gaussian representation and efficient rasterization. However, the 3DGS method notably exhibits performance degradation (average PSNR of 20 videos: 24.8), as it is tailored for static scenes and lacks design considerations for dynamic video scenes. On the other hand, the state-of-the-art baseline Deformable-3DGS, which employs a deformation field for time dimension on 3DGS, shows improved video reconstruction quality (average PSNR of 20 videos: 30.6); nevertheless, this enhancement comes at the expense of compromised training efficiency (50 minutes in average of 20 videos). Furthermore, due to the fundamental issue in COLMAP, they cannot conduct reconstruction on 8 videos, which hinders 3DGS from being used for wild video datasets. On the other hand, powered by the proposed MC-COLMAP and framework of Frg-3DGS/Bkg-3DGS, Video-3DGS can reconstruct all 28 videos in high quality with shorter training time (*iteration 3k*: **37.6 PSNR** with **11 minutes**; *iteration 5k*: **41.2 PSNR** with **22 minutes**; *iteration 10k*: **45.8 PSNR** with **56 minutes**; all results are measured by taking average over 28 videos). Upon training for 3k iterations, Video-3DGS significantly surpasses both the NeRF-based SoTA, Nerv (Chen et al., 2021b), and the 3DGS-based SoTA, Deform-3DGS (Yang et al., 2023), in terms of video reconstruction quality and training efficiency. Specifically, it achieves an improvement in PSNR by +3 and +7 over Nerv and Deform-3DGS, respectively. Furthermore, Video-3DGS demonstrates a notable improvement in training time efficiency, being 1.9 times faster than Nerv and 4.5 times faster than Deform-3DGS.

## 3.2 Video Editing

**Importance of 3D Gaussians for Video Editing Task** The primary goal of Video-3DGS is to enhance the temporal consistency of video editing by leveraging 3D Gaussian representations with a two-stage approach. Specifically, in the 1st stage, our Video-3DGS reconstructs videos using 3D Gaussians to establish temporal correspondence between frames within each clip and link neighboring clips with overlapping frames. It

motivates us to prove the video reconstruction capability of Video-3DGS, showcasing its ability to preserve the content and structure of original raw video without need of any 3D task exploration. Afterwards, we are able to exploit this reconstruction ability for video editing tasks in the 2nd stage. We provide explicit evidence of 3D Gaussians' suitability for temporal correspondence as shown in this video link, where we visualize the correspondence for the first five groups of 3D Gaussians. As shown in the visualization, Video-3DGS ensures the correspondences of similar areas across frames using 3D Gaussians of the same group. We can enhance this further by incorporating the visualization of pixel trajectories tracked by the 3D Gaussians. This enhancement is demonstrated in the provided video link.

**Baselines** We meticulously choose three zero-shot video editing methods as baseline comparisons. 1) Text2Video-Zero (Khachatryan et al., 2023), which extends Instruct-pix2pix (Brooks et al., 2023) to the video domain by inserting temporal attention within the diffusion model. 2) TokenFlow (Geyer et al., 2023), which achieves temporal smoothness through the propagation of diffusion features using inter-frame correspondences. 3) RAVE (Kara et al., 2023), which employs a noise-shuffling strategy and grid trick for enhancing video editing capabilities.

**Quantitative Results** We assessed Video-3DGS's editing capability on top of those three zero-shot video editors in Table 2. Video-3DGS significantly enhances temporal consistency (WarpSSIM score) across various editing scenarios spanning three different datasets. Consequently, this improvement generally yields superior final video editing results ($Q_{edit}$). It proves that Video-3DGS generally provides robust temporal editing guidance to existing video editors. We also observe that our proposed recursive and ensembled refinement can further improve both temporal consistency and video editing results on all of video editors and datasets.

**User Study** To explore user preferences for video editing outcomes, we conducted a user study detailed in Figure 6. We enlisted 20 participants and presented them with 30 randomly paired videos (10 pairs for each of the adopted off-the-shelf video editors) derived from the LOVEU video results. Participants were tasked with selecting their preferred edited video, taking into account fidelity to the provided text prompt (for editing) and temporal consistency. Consistent with the findings in the left panel of Figure 6, we observed that users generally favored Video-3DGS guided video editing results over the original edited videos. Moreover, in the right panel of Figure 6, we conducted another user study to show the effectiveness of recursive and ensembled refinement in editing with Video-3DGS.

## 4 Discussion & Limitations

While the proposed Video-3DGS exhibits remarkable performance in both video reconstruction and editing tasks, certain challenges persist.

In video reconstruction, our Video-3DGS still relies on the success of MC-COLMAP even though it is more robust version compared to original COLMAP. Following are scenarios when MC-COLMAP can struggle or fail. *Significant Occlusions of Foreground Objects*: When a foreground object becomes fully or partially occluded in consecutive frames, MC-COLMAP's decomposition process can struggle to track and re-identify the object. *Excessive Motion Blur*: Rapid camera or object movements can induce motion blur, making it difficult for feature detection algorithms to match points across frames accurately. If the blurred regions prevent consistent feature extraction, the reconstruction pipeline may fail to build a coherent model for certain frames or produce large alignment errors. We further acknowledge that MC-COLMAP for Video-3DGS requires a preprocessing step for foreground object extraction using an off-the-shelf panoptic segmenter. However, this is notably simpler compared to the five-stage preprocessing pipeline of shape-of-motion Wang et al. (2024a), which involves mask estimation, metric depth, monocular depth, camera estimation, and 2D tracking. Furthermore, we studied the influence of different variants and settings of the off-the-shelf segmenter on video reconstruction quality to ensure robustness as we discuss more in "Ablations on Segmentation Modules" of Appendix D.1.

In video editing, unsatisfactory results may arise if the underlying video editor fails or if significant changes to object shapes are required, which contradicts the inherent property of 3D Gaussians prioritizing the preservation of original structure. As a future direction for Video-3DGS, we aim to develop an architecture capable of manipulating 3D Gaussians in a more flexible manner to accommodate desired shape changes. Nevertheless, our work represents a pioneering effort in extending 3DGS to enhance the temporal consistency

of initially edited videos from any zero-shot video editor. We envision that this will inspire further research in a zero-shot video editing community.

Unlike other conventional neural representation papers, our work does not explore the task of novel view synthesis. Instead, our method focuses on video reconstruction and editing by encoding and fitting videos with 3DGS. We anticipate that our Video-3DGS will also serve as a fundamental framework for 4D novel view synthesis in future research.

## 5 Conclusion

We introduced **Video-3DGS**, a novel two-stage approach tailored to edit dynamic monocular video scenes by reconstructing them with 3D Gaussian Splatting (3DGS). The first stage of Video-3DGS lies its capacity to spatially and temporally decompose videos, streamlining motion representation across clips. It surpasses previous state-of-the-art methods in video reconstruction. The second stage of Video-3DGS leverages the reconstruction ability to refine the edited videos from three different zero-shot video editors, which enhances both temporal consistency and editing results. We anticipate significant positive societal impacts from our method, as it consistently performs well in representing diverse videos and refining outputs from zero-shot video editors. We expect it to benefit various applications, such as Entertainment with video synthesis and AR/VR.

**Acknowledgement** This work was supported in part by the Institute of Information and Communications Technology Planning and Evaluation (IITP) grant funded by the Korea Government (MSIT) (Artificial Intelligence Innovation Hub) under Grant 2021-0-02068. This research was also partially supported by the KAIST Cross-Generation Collaborative Lab Project.

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

## Appendix

In the appendix, we provide additional information as listed below:

- Appendix A provides the related works.

- Appendix B provides the details of used datasets and metrics.

- Appendix C provides the implementation details.

- Appendix D provides extensive ablation study on the proposed Video-3DGS.

- Appendix E provides qualitative results on video reconstruction and video editing.

- Appendix F provides the licenses of the assets used.

- Appendix G discusses broader impact.

## A  Related Work

In this part, we introduce the literature related to the core part of Video-3DGS (1st stage): neural rendering for scene representation (Appendix A.1), 3D Gaussian Spatting-based representation (Appendix A.2), dynamic scene representation (Appendix A.3), and spatial temporal decomposition in neural rendering (Appendix A.4). Regarding to Video-3DGS (2nd stage), we briefly review the literature on Text-to-Image editing (Appendix A.5), and Zero-shot Text-to-Video editing (Appendix A.6).

### A.1  Neural Rendering for Scene Representation

The neural radiance field (NeRF) (Mildenhall et al., 2021) is one of the advanced methodologies that represent scenes using implicit neural rendering with MLP. It has seen significant advancement recently, with major improvements in several key areas. Firstly, the speed of both training (Karnewar et al., 2022) and inference (Garbin et al., 2021) has been substantially increased, allowing the current models to be trained and deployed within minutes (Barron et al., 2021; Yu et al., 2021; Reiser et al., 2021; Martin-Brualla et al., 2021). Further advancements have addressed image artifacts, including motion blur (Ma et al., 2022), exposure and lens distortion (Jeong et al., 2021), enhancing the quality and realism of rendered scenes. The necessity for precise camera calibrations is becoming less stringent, thanks to new methods that allow for local camera refinement (Chng et al., 2022; Lin et al., 2021). More recent research has also extended NeRF applications beyond traditional uses, exploring areas such as generalization (Wang et al., 2022; Chen et al., 2021a), semantic understanding (Zhi et al., 2021; Kundu et al., 2022), robotics (Zhou et al., 2023), and 3D image style transfer (Gu et al., 2021). These developments indicate a broadening scope and increasing versatility of NeRF models in capturing and rendering complex scenes.

### A.2  3D Gaussian Spatting-based Representation

Building upon the advancements in Neural Radiance Fields (NeRF) (Kerbl et al., 2023) for scene representation, another innovative approach that has gained attention is the utilization of 3D Gaussian distributions to model scenes. This methodology, often referred to as "Gaussian Scene Representation", leverages the mathematical properties of Gaussian distributions to efficiently capture the volumetric density and color information of 3D scenes. Similar to the trend in NeRF, there have been several advancements in performance and efficiency. For example, some of them apply the module used in NeRF to 3D Gaussian-based view synthesis. (Lee et al., 2024) leverages a compact MLP to adjust the covariance of 3D Gaussians, which can effectively reconstruct sharp details from blurry images and Mip-splatting (Yu et al., 2023) address alias issue in 3DGS as similar to (Barron et al., 2021). Besides the advancements in quality, (Fan et al., 2023) enhances the efficiency and compactness of 3D Gaussian Splatting for neural rendering by pruning insignificant Gaussians.

### A.3 Handling Dynamic Scene Representation

Despite the significant progress in NeRF and 3DGS for static scene representation, both approaches encounter substantial challenges when applied to dynamic scenes. The core limitation stems from their inherent design, which is primarily tailored for static environments, thus struggling to adapt to changes in scene geometry and appearance over time. To address these challenges, recent research endeavors have begun exploring extensions and alternatives to NeRF and 3D Gaussian Splatting that incorporate temporal information into the scene modeling process. For instance, deformation network has been widely used for encoding time dimension both in the framework of NeRF and 3DGS, (Fan et al., 2023) and (Yang et al., 2023) respectively. More developments for dynamic scene representation are introduced in NeRF. Robust-Dyn (Liu et al., 2023) enhances the robustness of dynamic radiance field reconstruction by jointly estimating static and dynamic scene components along with camera parameters. NeRV Chen et al. (2021b) utilizes neural implicit representation to encode videos by simply fitting a neural network to input video frames. Yet, despite their advancements, due to the fundamental limitation in the NeRF framework, these approaches continue to grapple with challenges related to achieving fine detail, maintaining temporal consistency across frames, and requiring extensive optimization times. Driven by these challenges, the proposed Video-3DGS seeks to investigate the application of 3DGS for dynamic scenes captured through monocular video, exploring its efficacy and potential improvements.

### A.4 Spatial Temporal Decomposition

Our proposed Video-3DGS decomposes the dynamic video spatially and temporally to represent original video effectively. Similarly, ProgressiveNeRF (Meuleman et al., 2023) proposes method of decomposing video into spatial and temporal components through progressively optimized local radiance fields. It represents a significant advancement in the field of robust view synthesis and large-scale scene reconstruction. The primary distinction between our method and the ProgressiveNeRF approach lies in how we manage camera poses and frame integration across video clips. While ProgressiveNeRF dynamically updates camera poses and continues to append frames until the camera's pose extends to the boundary of a high-resolution, uncontracted space, our method treats each clip independently. We utilize overlapping frames solely to ensure consistency of 3D Gaussians corresponding to overlapping frames across neighboring clips, with each clip maintaining its unique camera poses. However, all clips can still be rendered onto the same image plane via a differentiable rendering with each distinctive 3D Gaussian and camera pose. Additionally, our process repeats until MC-COLMAP signals a 'Success' *status* marking a clear difference in the criteria used to determine the sufficiency of frames for each clip. Furthermore, ProgressiveNeRF uses several dependencies (*e.g.*, depth estimation and optical flow) and many losses for better temporal decomposition. However, we only need segmentation masks and simple reconstruction losses (L1 and SSIM) as in the original 3DGS paper. We also provide extensive visual comparison with ProgressiveNeRF in this link.

### A.5 Text-to-Image Editing

Text-to-image editing has significantly advanced with the development of generative models, particularly latent diffusion models (Rombach et al., 2022). These models have demonstrated remarkable capabilities in generating high-quality images guided by textual prompts. Methods such as DreamBooth (Ruiz et al., 2022) and Textual Inversion (Gal et al., 2022) have achieved notable success by fine-tuning pre-trained models for specific editing tasks. Recent approaches, like Prompt-to-Prompt (Hertz et al., 2022) and DiffEdit (Couairon et al., 2022), leverage attention mechanisms within these models to enable localized and detailed image edits without extensive retraining. Techniques like SDEdit (Meng et al., 2022) utilize stochastic differential equations to guide image synthesis, enabling precise control over the generated content. In addition to these, Blended Diffusion (Avrahami et al., 2022) presents a method to blend newly generated content seamlessly into existing images. This technique ensures that the synthesized additions are coherent with the original image context, thus maintaining a high level of realism. Another notable approach is Paint by Word (Andonian et al., 2021), which introduces a framework where users can interactively edit images by providing textual descriptions for specific regions. This method utilizes a combination of mask prediction and image generation

to accurately reflect the user's intent. Despite significant advancements in text-to-image editing, the progress in text-to-video editing technology has been relatively slow.

### A.6 Zero-shot Text-to-Video Editing

Zero-shot text-to-video editing leverages pre-trained text-to-image models to edit videos without requiring extensive retraining. This approach aims to overcome the limitations of traditional video editing methods, such as video-specific training (Ho et al., 2022) and atlas learning (Kasten et al., 2021), which are often time-consuming and require significant manual effort. Recent advancements in this domain have introduced various techniques to achieve high-quality and temporally consistent video edits. For instance, RAVE (Kara et al., 2023) employs a noise shuffling strategy to enhance temporal consistency by leveraging spatio-temporal interactions between frames. This method integrates spatial guidance through ControlNet (Zhang et al., 2023) and operates efficiently in terms of memory requirements, making it suitable for editing longer videos. RAVE's approach significantly reduces processing time compared to existing methods, achieving roughly 25% faster editing rate while maintaining high visual quality. Other notable zero-shot methods include Pix2Video (Ceylan et al., 2023) and FateZero (Qi et al., 2023), which utilize sparse-causal attention and attention feature preservation, respectively, to maintain motion and structural consistency across video frames. Text2Video-Zero (Khachatryan et al., 2023) synthesizes and edits videos by integrating cross-frame attention and controlling the fidelity to structure of original video with image guidance scale. TokenFlow (Geyer et al., 2023) enforces consistency by propagating diffusion features across frames based on inter-frame correspondences in zero-shot manner. These techniques collectively demonstrate the potential of zero-shot video editing in producing temporally coherent and visually appealing videos without the need for extensive training on video-specific datasets. Despite the recognized improvements in efficiency and editing capabilities offered by zero-shot video editors, they still suffer from subpar temporal consistency and overall video editing quality due to a lack of per-scene understanding. This limitation motivates us to design a plug-and-play refiner for zero-shot video editors (we select representative three editors: Text2Video-Zero / TokenFlow / RAVE), aimed at enhancing per-scene understanding and improving the quality of the edited videos.

## B Datasets and Metrics

### B.1 Datasets

**Video Reconstruction**  To assess the quality of video reconstruction in real-world monocular video scenes, we curated a dataset comprising 28 representative videos from the DAVIS dataset (Pont-Tuset et al., 2017), each with a resolution of 480p. These videos feature a varied array of foreground moving objects, encompassing humans, vehicles, and animals, together with cluttered backgrounds. Belows are the key features of DAVIS dataset:

- Resolution: videos are available in two resolutions, 480p and 1080p. Following prior works, we utilized 480p.

- Each video has a different number of frames ranging from 40 to 80

- FPS: 24

- Video sequences: the dataset includes 50 video sequences. Among them, we selected 28 videos that feature a varied array of foreground-moving objects, encompassing humans, vehicles, and animals.

**Video Editing**  For video editing, we adopt the datasets and text prompts used in the CVPR 2023 workshop, LOVEU Text-Guided Video Editing (TGVE) challenge (Wu et al., 2023b). They curated a dataset consisting of 76 videos (480p) in-the-wild from DAVIS, Youtube, and Videvo Videos. Each video is edited according to 4 different types of text prompts acquired from BLIP-2 (Li et al., 2023), including: style change, object change, background change, and multiple change. As noted in the main paper, we tested our framework in 58 videos with 4 editing prompts, totaling 232 video editing scenarios. Below are key details.

| COLMAP | Type | COLMAP | | | | Masked COLMAP | | MC COLMAP | |
|---|---|---|---|---|---|---|---|---|---|
| | success rate | 20/28 | | | | 25/28 | | 28/28 | |
| | processing time | 12m30s | | | | 2m10s | | 2m3s | |
| 3DGS | Type | Deformable-3DGS | Frg-3DGS | | | Frg-3DGS + Bkg-3DGS | | Frg-3DGS + Bkg-3DGS | |
| | iteration / training time | 40k (50m) | 10k (4m) | 30k (15m) | 60k (34m) | 10k (8m) | 30k (34m) | 3k (11m) | 5k (21m) |
| | PSNR | 30.6 | 32.9 | 35.3 | 37.3 | 35.1 | 38.0 | 37.6 | 41.2 |

Table 3: Ablation study on MC-COLMAP. The colored cells indicate the best, second best, and third best performances (best viewed in color). Our MC-COLMAP not only successfully generates 3D points for all 28 videos, but also enables the high reconstruction quality.

- Resolution: 480x480

- Frame number: DAVIS: 32 / Youtube: 128 / Videvo: 32

- FPS: 24

## B.2 Evaluation Metrics

**Video Reconstruction** To evaluate video reconstruction quality, two principal metrics are utilized: Peak Signal-to-Noise Ratio (PSNR) and Structural Similarity Index Measure (SSIM). These metrics collectively appraise the fidelity of the reconstructed video in comparison to the original. Additionally, efficiency is gauged through the metric of Training Time, which quantifies the required training time, thus indicating the computational efficiency of the reconstruction algorithm.

**Video Editing** To assess video editing quality, we utilize WarpSSIM, which computes the mean SSIM score between the edited video warped by the optical flow (Teed & Deng, 2020) (derived from the source video) and corresponding original edited video. This metric offers valuable insight into the temporal consistency of post-editing. Furthermore, we employ $Q_{edit}$ (Kara et al., 2023), a comprehensive video editing metric that combines the WarpSSIM score with the CLIPScore_text, providing a multiplicative assessment of the overall video editing performance.

## C  Implementation Details

Our framework is based on Pytorch (Paszke et al., 2019) and 3D Gaussian Splatting (Kerbl et al., 2023), while adopting depth incorporated differentiable Gaussian rasterization (Yang et al., 2023). During training, two sets of 3D Gaussians (both Frg-3DGS and Bkg-3DGS) in each clip are sequentially optimized with three different total numbers of training iterations: a total of 3k, 5k and 10k iterations, each targeting a distinct time-accuracy trade-off (*e.g.*, 10k iterations for the best quality but the slowest training time). We employ the same training hyper-parameters (*e.g.*, learning rate, deformation network setting) as specified by (Yang et al., 2023). With optimized 3D Gaussians for each clip, video reconstruction proceeds through sequential differential rendering (Kerbl et al., 2023) of the clips. For video editing, 3D Gaussians optimized for original videos undergo fine-tuning for 1k iterations to update the spherical harmonic coefficients (SH) and opacity ($\sigma$), guided by the initial edited videos (obtained from an off-the-shelf zero-shot video editor). Each experiment uses a single A100 GPU.

## D  Ablation Study

### D.1  Video-3DGS (1st Stage)

**Analysis on MC-COLMAP** We present a comprehensive analysis of MC-COLMAP. Table 3 compares two baselines: conventional COLMAP, which processes full video frames, and Masked COLMAP, which employs only spatial decomposition. Conventional COLMAP encounters more failure case (8 out of 28 videos fail to obtain 3D points) and exhibits slower processing time (12m30s). Among the successful cases (20 videos), reconstruction using the state-of-the-art Deformable-3DGS (Yang et al., 2023) performs suboptimally in

| Scene | blackswan | boat | drift-turn | horsejump-high |
|---|---|---|---|---|
| Clip #1 | 17(1) | 11(1) | 10(1) | 10(1) |
| Clip #2 | 11(1) | 13(1) | 11(1) | 11(1) |
| Clip #3 | 11(1) | 17(1) | 11(1) | 11(1) |
| Clip #4 | 11(1) | 20(5) | 11(1) | 11(1) |
| Clip #5 | 4 | 22 | 11(1) | 11 |
| Clip #6 | - | - | 11(1) | - |
| Clip #7 | - | - | 8 | - |

Table 4: Analysis on statistics of MC-COLMAP. We show the number of frames for each clip in different scenes. The number in the parentheses refers to the number of overlapping frames between neighboring clips.

| | Metrics | PSNR | SSIM |
|---|---|---|---|
| Method | | | |
| Frg-3DGS (w/ fore) | | 35.1 | 0.923 |
| Frg-3DGS (w/ rdn) | | 36.5 | 0.951 |
| Frg-3DGS (w/ fore+rdn) | | 37.5 | 0.957 |
| Frg-3DGS + Bkg-3DGS | | 41.4 | 0.987 |

Table 5: Analysis on Frg-3DGS and Bkg-3DGS. 'fore' and 'rnd' denote the foreground and random 3D points, respectively. Utilizing a single set of Frg-3DGS can benefit from both types of 3D points. However, performance can be enhanced further by employing two sets of Frg-3DGS and Bkg-3DGS, tailored specifically for foreground and background random 3D points, respectively.

both training time and PSNR compared to our Frg-3DGS, powered by multi-resolution hash encoding-based deformation network. It is worth noting that this study utilizes only one set of 3D Gaussians, without spatial decomposition for foreground and background points. Masked COLMAP demonstrates improved success rate (only 3 out of 28 videos fail to obtain 3D points) and enhances the processing speed by mitigating the effects of cluttered backgrounds through spatial decomposition. Additionally, reconstruction quality, as measured by PSNR, is improved. Finally, our proposed MC-COLMAP, which incorporates both spatial and temporal decomposition, successfully captures 3D points from all 28 videos. Furthermore, it enables the best reconstruction quality of 41.2 PSNR with 5k training iterations, significantly outperforming the other settings.

**MC-COLMAP Statistics**  As shown in Table 4, to enhance comprehension of MC-COLMAP, we present statistics that illustrate the distribution of clip length across four representative scenes from the challenging DAVIS dataset. We have selected a default value of k=10 for our experiments and found out that altering k to either 5 or 15 does not markedly impact our results. We synchronize different 3DGS sets across video clips through the use of overlapping frames, which are indicated in the above table as numbers within parentheses. Initially, we designate the count of overlapping frames to be one, although this number can be adjusted based on the number of frames added to each clip. It also shows behaviors for slow or fast motion in video. For example, the scenes "blackswan" and "boat" feature comparatively static and slow movements, whereas "drift-turn" and "horsejump-high" are characterized by highly dynamic actions. Each scene begins with an initial set of 10 frames for MC-COLMAP processing. It is noted that fast-moving scenes are typically decomposed into 10 or 11 frames. In contrast, scenes with slower motion demand a greater number of frames to accurately extract 3D points for MC-COLMAP analysis.

**Ablations on Segmentation Modules**  we additionally include the ablation study on using different video segmentation modules. There are several open-vocabulary video object segmentation models available, and we have chosen the most representative models for our experiments, including DEVA model with SAM, and Mobile-SAM. We experimented with three settings: (a) DEVA with SAM and user prompts (i.e., user needs to provide the object class names), (b) DEVA with SAM and VIPSeg foreground categories (i.e., we use the foreground categories defined by VIPSeg dataset without any user prompts), and (c) DEVA with Mobile-SAM and VIPSeg foreground categories. As shown in this video, our Video-3DGS is robust

| DAVIS (480p) | PSNR | SSIM | WarpSSIM | Qedit |
|---|---|---|---|---|
| CoDeF | 28.7 | 0.904 | 0.926 | 23.7 |
| Video-3DGS (3k) | 40.8 | 0.986 | - | - |
| CoDeF+Video-3DGS | - | - | 0.901 | 23.2 |
| RAVE | - | - | 0.872 | 23.4 |
| RAVE+Video-3DGS | - | - | 0.908 | 24.1 |
| RAVE+Video-3DGS (RE) | - | - | 0.913 | 24.8 |

| Youtube | PSNR | SSIM | WarpSSIM | Qedit |
|---|---|---|---|---|
| CoDeF | 25.5 | 0.795 | 0.925 | 21.8 |
| Video-3DGS (3k) | 40.8 | 0.986 | - | - |
| CoDeF+Video-3DGS | - | - | 0.924 | 21.2 |
| TokenFlow | - | - | 0.848 | 22.1 |
| TokenFlow+Video-3DGS | - | - | 0.923 | 23.1 |
| TokenFlow+Video-3DGS (RE) | - | - | 0.923 | 24.2 |

Table 6: Comparison of the proposed Video-3DGS method with the CoDEF Ouyang et al. (2023) framework in the context of video editing tasks. The results demonstrate that Video-3DGS exhibits strong compatibility with zero-shot video editors, yielding superior or comparable performance to CoDEF across several video editing benchmarks.

to the change of segmentation modules. Furthermore, it can also prove that our MC-COLMAP effectively reconstructs foreground moving objects regardless of their predefined categories, thereby achieving comparable reconstruction capabilities.

**Analysis on Frg-3DGS and Bkg-3DGS** In Table 5, we demonstrate the necessity of employing two sets of Frg-3DGS and Bkg-3DGS for video reconstruction. We begin with a single set of Frg-3DGS that leverages only the foreground 3D points (denoted as Frg-3DGS w/ fore), resulting in 35.1 PSNR and 0.923 SSIM. By initializing the single set of Frg-3DGS with the spherical-shaped random points (denoted as Frg-3DGS w/ rdn), the performance improves to 36.5 PSNR and 0.951 SSIM. The inclusion of random 3D points uniformly captures both foreground and background points, thus enhancing performance. Furthermore, employing both foreground 3D points and spherical-shaped random 3D points (Frg-3DGS w/ fore+rdn) yields further improvement. Finally, utilizing two sets of Gaussians, Frg-3DGS and Bkg-3DGS, each specific to foreground and background 3D points respectively, significantly enhances performance to 41.4 PSNR and 0.987 SSIM.

**2D Learnable Parameter for Merging Foreground and Background Views** We visualized the 2D learnable parameter (*i.e.*, $\alpha$ in Equation (5)) in different video scenarios in the following link. As shown in the videos, the learned $\alpha$ focuses on the foreground regions, while the learned $(1 - \alpha)$ focuses on the background regions.

**Model Size** The overall size of the model is determined by both the number of clips and the size of the Frg-3DGS and Bkg-3DGS for each clip. For example, consider the "blackswan" scene. the Video-3DGS (3k iteration) necessitates, on average, a 15MB Frg-3DGS and a 60MB Bkg-3DGS per clip. Consequently, the cumulative model size amounts to 375MB. In contrast, the conventional 3DGS (30k iteration) employs a singular, substantial 3DGS model, approximately 300MB in size, which is a result of the densification technique employed during long iterations. Despite the relatively big model size required for 3DGS, since it represents and stores the scene in an explicit manner (point cloud), we can take advantage of the efficient training and inference capability of 3DGS in videos. It should be noted that the aggregate size of the Video-3DGS model escalates with an increasing number of clips. Therefore, a pivotal objective for future research is to minimize the overall size requisite for Video-3DGS implementation. One of the naive ways of achieving this is opting for a larger k value so that we can effectively reduce the final model size, as this leads to a decreased number of clips.

## D.2 Video-3DGS (2nd Stage)

**Comparison With NeRF-based Video Editing** In addition to Table 2 where we show the compatibility of Video-3DGS with zero-shot video editors, we compare with a training-based video editor that utilizes NeRF Mildenhall et al. (2021) CoDEF Ouyang et al. (2023) as in Table 6. We can observe that Video-3DGS with zero-shot video editors (e.g., RAVE or TokenFlow) show comparable or better results than "CoDEF+Video-3DGS" in terms of both reconstruction and editing ability. Qualitative Results with CoDEF can be found in this link. As shown in the qualitative results, CoDEF struggles to preserve the content and structure of the original video due to its application of ControlNet on a canonical image. In contrast, our Video-3DGS effectively maintains the original content and structure.

| Metrics 
 Method | WarpSSIM↑ | $Q_{edit}$↑ |
|---|---|---|
| UniEdit (Bai et al., 2024) | 0.779 | 19.95 |
| +Video-3DGS | 0.830 | 21.15 |
| AnyV2V (Ku et al., 2024) | 0.737 | 19.85 |
| +Video-3DGS | 0.836 | 22.25 |

Table 7: Additional results on compatibility of Video-3DGS with video editors using video generation models. Our Video-3DGS shows consistent compatibility with video editors using video generation model by increasing the editing quality. We used 16 DAVIS videos with 32 frames and initially edited them with style-change prompts from LOVEU (Wu et al., 2023b) benchmark.

| | Text2Video-Zero | + Video-3DGS | TokenFlow | + Video-3DGS | RAVE | +Video-3DGS |
|---|---|---|---|---|---|---|
| Editing time | 1m11 | +6m7s | 2m50s | +6m7s | 7m20s | +6m5s |

Table 8: Ablation study on time cost of Video-3DGS ('style change' in DAVIS).

**Comparison With Editors Using Video Generation Models** Recently, a new trend in video editing has emerged by leveraging video generation models. These approaches can be broadly categorized into two types: those that use image-to-video (I2V) priors and those that employ text-to-video (T2V) priors. To illustrate, we selected a representative method from each category—UniEdit (Bai et al., 2024) (T2V-based) and AnyV2V (Ku et al., 2024) (I2V-based). Due to UniEdit's high memory requirements, we divided DAVIS videos (32 frames each) into four chunks and applied the LaVie (Wang et al., 2024b) T2V model to each chunk as a prior for video editing. For AnyV2V, we edited the first frame using Instruct-pix2pix (Brooks et al., 2023) and then propagated the edits to the remaining frames using its I2V model. After obtaining the initial outputs from each methods, we refined the results with our Video-3DGS. As shown in Table 7, Video-3DGS demonstrates strong compatibility with recent methods that use video generation priors, particularly when processing DAVIS videos with style-change prompts from the LOVEU benchmark.

**Time Cost Analysis** Similar to Table 1, we also explore the time efficiency of video editing when incorporating Video-3DGS atop existing zero-shot video editing frameworks in this section in Table 8. This assessment spans three critical phases: optimizing Frg-3DGS and Bkg-3DGS with the initial video, updating $SH$ and $\sigma$ for the initially edited video, and rendering to produce a refined edit. We can demonstrate that integrating Video-3DGS as an enhancement module consistently requires approximately 6 minutes, irrespective of the underlying editing techniques, affirming its efficiency as a reasonable investment in processing time.

**Analysis on Recursive and Ensembled Refinement** After observing the effectiveness of Video-3DGS (RE) in Table 2, we conducted an ablation study on its components, as shown in Table 9. Each component individually improves performance compared to a single-phase refiner. The 'Recursive' component increases WarpSSIM by 1.5% and $Q_{edit}$ by 0.4%. Similarly, the 'Ensembled' component enhances the refiner, resulting in a 7% increase in WarpSSIM and a 0.4% increase in $Q_{edit}$. Our Video-3DGS (RE), which combines both recursive and ensembled strategies, achieves the highest scores in both WarpSSIM (**0.899**) and $Q_{edit}$ (**22.3**).

### D.3 Relationship between Video-3DGS (1st Stage) and Video-3DGS (2nd Stage)

We examine the correlation between video reconstruction and editing quality in Table 10. We introduce two versions of Video-3DGS: model A, where all clips utilize the same deformation field (thus reducing the representation capacity), and model B, our default setting. As depicted in the table, model B achieves superior reconstruction quality compared to model A, resulting in better performance across all three off-the-shelf video editors. We also conducted qualitative comparison with 3DGS (Kerbl et al., 2023) to further prove the positive relationship between video reconstruction and video editing in this link.

| Dataset | Method | Text2Vid-Zero | | +Video-3DGS | | WarpSSIM↑ | $Q_{edit}$↑ |
|---|---|---|---|---|---|---|---|
| | | WarpSSIM↑ | $Q_{edit}$↑ | Recursive | Ensembled | | |
| DAVIS | | 0.691 | 20.1 | | | 0.827 | 21.0 |
| | | | | ✓ | | 0.842 | 21.4 |
| | | | | | ✓ | 0.897 | 21.4 |
| | | | | ✓ | ✓ | **0.899** | **22.3** |

Table 9: Analysis on the components of recursive and ensembled Video-3DGS.

| Text2Video-Zero | +Video-3DGS | | | TokenFlow | +Video-3DGS | | | RAVE | +Video-3DGS | | |
|---|---|---|---|---|---|---|---|---|---|---|---|
| Editing | Reconstruction | | Editing | Editing | Reconstruction | | Editing | Editing | Reconstruction | | Editing |
| WarpSSIM / Qedit | Model | PSNR | WarpSSIM / Qedit | WarpSSIM / Qedit | Model | PSNR | WarpSSIM / Qedit | WarpSSIM / Qedit | Model | PSNR | WarpSSIM / Qedit |
| 0.791 / 21.1 | A | 31.1 | 0.842 / 21.7 | 0.869 / 23.2 | A | 31.1 | 0.884 / 22.9 | 0.874 / 24.8 | A | 31.1 | 0.871 / 23.3 |
| | B | 41.3 | 0.841 / 22.8 | | B | 41.3 | 0.916 / 24.4 | | B | 41.3 | 0.905 / 25.6 |

Table 10: The relationship between video reconstruction and video editing on DAVIS. Across all three off-the-shelf video editors (Text2Video-Zero, TokenFlow, and RAVE), model B (with better reconstruction ability) yields better video editing results than model A.

# E Qualitative Results

In this section, we provide more qualitative results for both video reconstruction (Appendix E.1) and video editing (Appendix E.2).

## E.1 Video Reconstruction

For video reconstruction, we visualize qualitative results in Figure 7, Figure 8, and Figure 9. As shown in the figures, the proposed Video-3DGS consistently demonstrates higher reconstruction quality. Video comparisons with 3DGS (Kerbl et al., 2023) and Deformable-3DGS (Yang et al., 2023) are provided in the following link

### E.1.1 Spatial Decomposition

We also provide visual analysis on the spatial decomposition in the following video link. Our method successfully extracts the corresponding 3D points of moving objects using MC-COLMAP. We also found that the spatial decomposition can work well in a scene with multiple foreground objects.

## E.2 Video Editing

For video editing, we provide the visualization results for the 'Single-phase Refiner' and 'Recursive and Ensembled Refiner'.

### E.2.1 Single-phase Refiner

To show the visual effectiveness of Video-3DGS as single-phase refiner, we visualize qualitative results in Figure 10, Figure 11, and Figure 12, which adopt Text2Video-Zero (Khachatryan et al., 2023), TokenFlow (Geyer et al., 2023), and RAVE (Kara et al., 2023) as the underlying video editor, respectively. As shown in the figures, Video-3DGS effectively enhances the temporal consistency in the edited results across all three video editors. We further provide the video analysis of Video-3DGS as a plug-and-play refiner to show better visual comparison in this link.

### E.2.2 Recursive and Ensembled Refiner

We further conduct the visual analysis on Video-3DGS as Recursive and Ensembled (RE) refiner. First, similar to Figure 3, we provide more video results showing the sensitivity of video editors to hyperparameter changes in this link. It confirms that different hyperparameter values in the deployed video editors results in different outputs. Then, we show the editing visual improvement brought by Video-3DGS (RE) in here.

## F  Asset Licenses

The licenses of the assets used in the experiments are denoted as follows:

Datasets:

- DAVIS2017 (Pont-Tuset et al., 2017): https://davischallenge.org/index.html

- LOVEU-TGVE-2023 (Wu et al., 2023b): https://github.com/showlab/loveu-tgve-2023

Codes:

- 3D Gaussian Splatting (Kerbl et al., 2023): https://github.com/graphdeco-inria/gaussian-splatting

- Text2Video-Zero (Khachatryan et al., 2023): https://github.com/Picsart-AI-Research/Text2Video-Zero

- TokenFlow (Geyer et al., 2023): https://github.com/omerbt/TokenFlow

- RAVE (Kara et al., 2023): https://github.com/RehgLab/RAVE

Models:

- Stable Diffusion v 1.5 Models (Rombach et al., 2022): https://huggingface.co/runwayml/stable-diffusion-v1-5

## G  Broader Impact

This paper presents Video-3DGS, which leverages 3D Gaussian Splatting to reconstruct and edit dynamic monocular videos. We anticipate significant positive societal impacts from our method, as it consistently performs well in representing diverse videos and refining outputs from zero-shot video editors. We expect it to benefit various applications, such as Entertainment with video synthesis and AR/VR.

GT video

Figure 7: Qualitative results for video reconstruction.

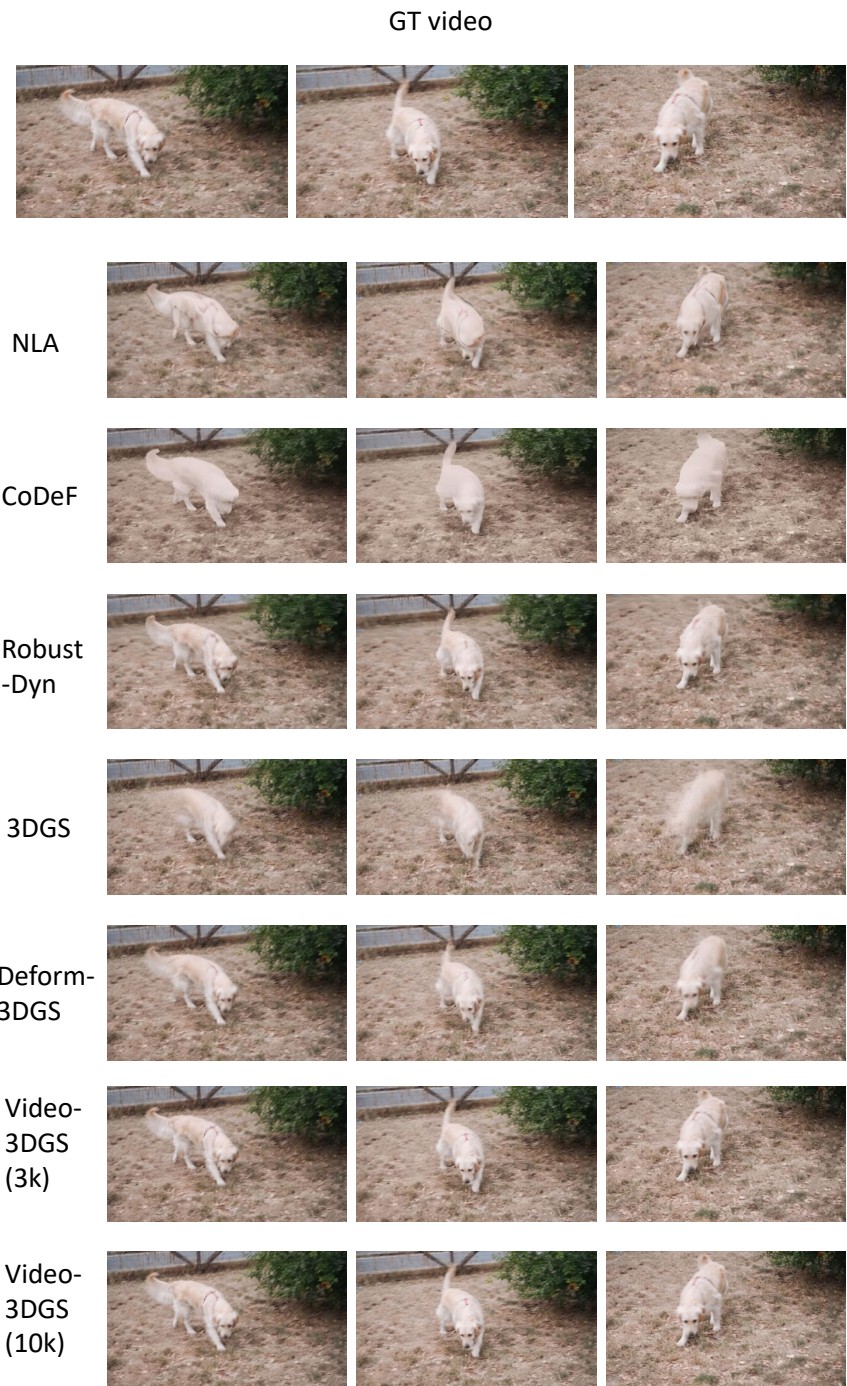

Figure 8: Qualitative results for video reconstruction.

GT video

Figure 9: Qualitative results for video reconstruction.

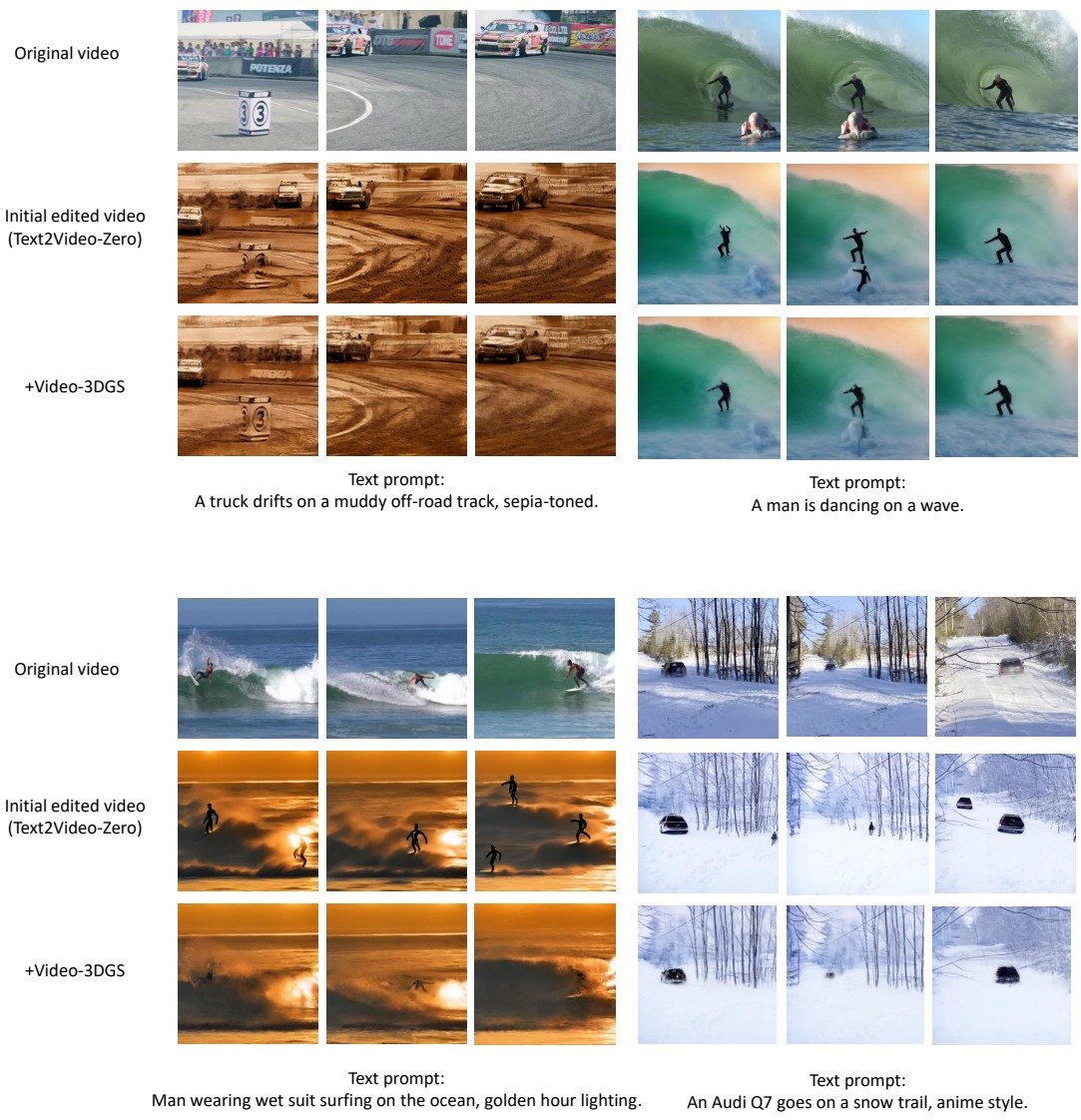

Figure 10: Qualitative results of single stage refiner for video editing by adopting Text2Video-Zero.

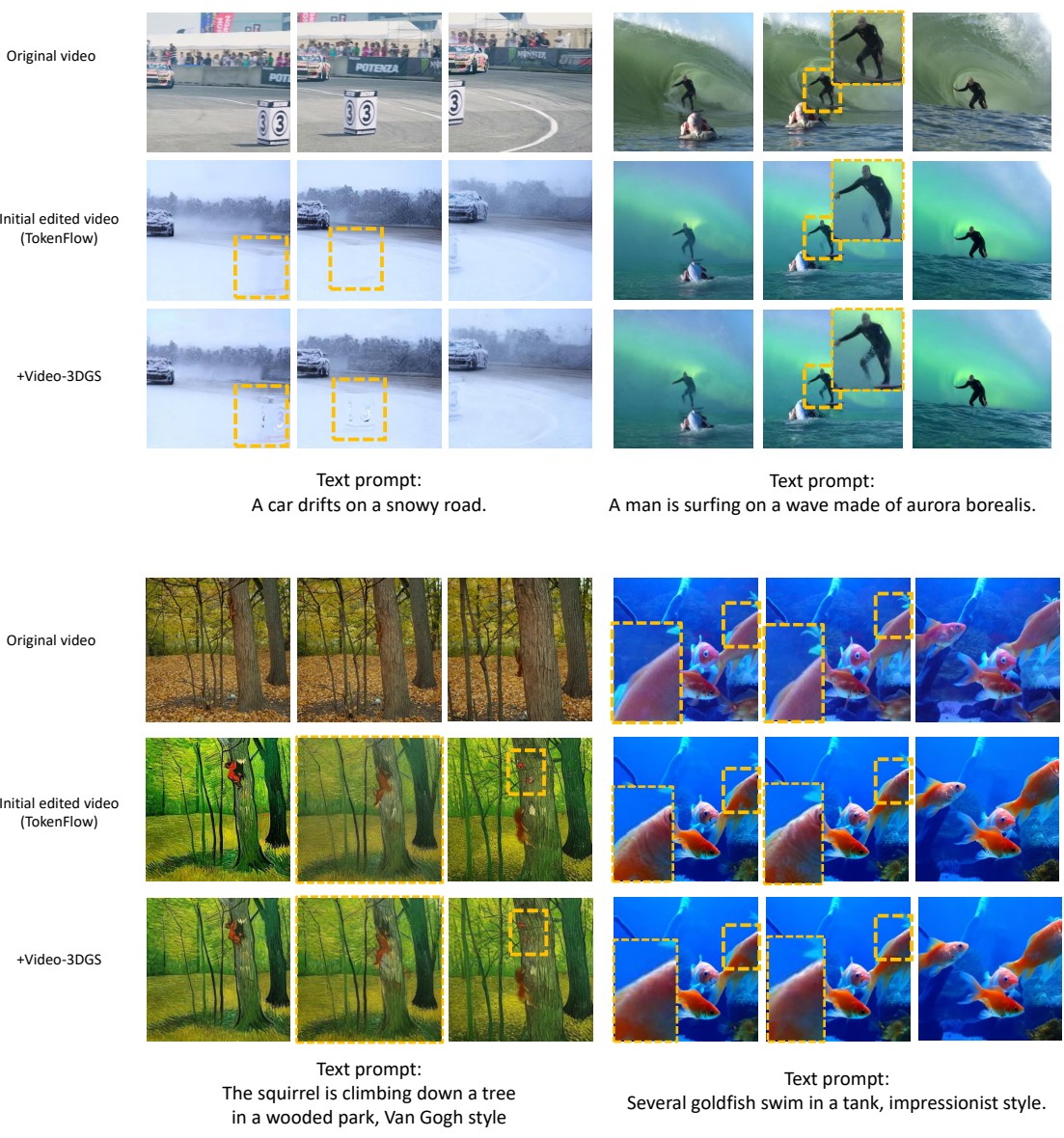

Figure 11: Qualitative results of single stage refiner video editing by adopting TokenFlow.

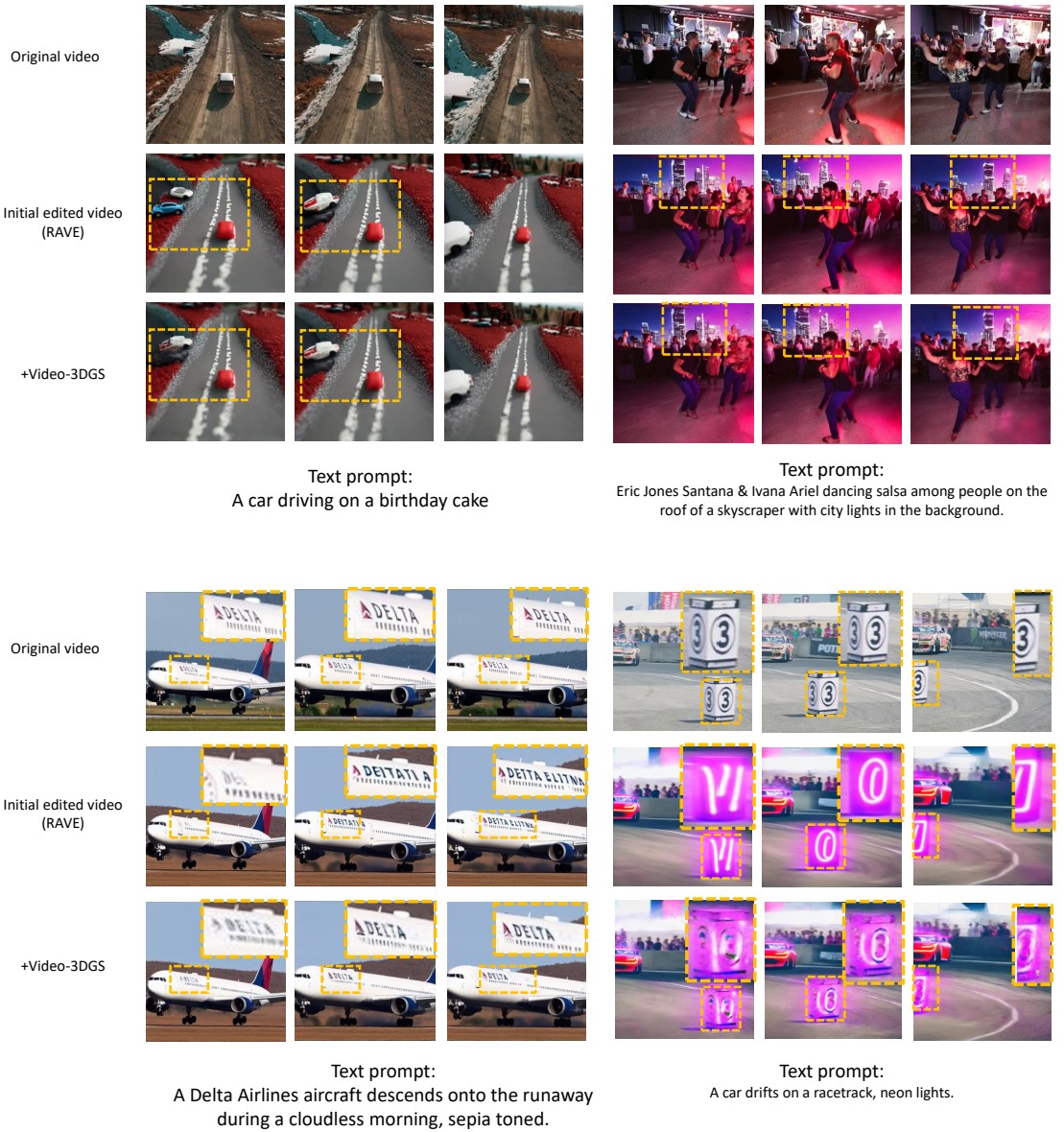

Figure 12: Qualitative results of single stage refiner for video editing by adopting RAVE.

