# OpenReview forum: "Enhancing Temporal Consistency in Video Editing by Reconstructing Videos with 3D Gaussian Splatting"
_TMLR — Accepted by TMLR_

### Review · Reviewer_274s · 2025-01-02

**Summary Of Contributions:**

- **Novel pipeline for video editing** based on 3DGS and a diffusion model.
- The method ensures **temporal consistency** by introducing 3DGS-based explicit scene modeling and a zero-shot video style change using an off-the-shelf video diffusion model.
- **Experimental results** demonstrate **competitive or improved performance** on benchmark datasets.

**Audience:**

No

**Broader Impact Concerns:**

Nothing

**Claims And Evidence:**

Yes

**Requested Changes:**

Because the **technical contribution of the work is fundamentally marginal**, a more extensive evaluation is necessary to validate the practical usefulness of the method. In particular, please include comparisons with:

- **Dynamic NeRF/3DGS methods** such as RodyNeRF and Shape-of-Motion.
- **Other video editing methods** such as InstructNeRF2NeRF and Gaussian Editor.

**Strengths And Weaknesses:**

**Strengths:**

- **Practical system design** that decomposes the problem into a multi-stage optimization process.
- **Improved performance** on benchmark datasets.

**Weaknesses:**

- The proposed method **primarily focuses on system integration**, resulting in limited technical novelty. Each technical component (MC-COLMAP, Deformable GS, video editing) is essentially a straightforward extension or hyperparameter tuning, and does not offer substantial new insights.
- The work is **more of an engineering case study** and may not be ideally suited for an ML-focused venue; it might be better positioned in a more applied CV venue.

---

> ### Author Response · Authors · 2025-01-10
> **To Reviewer 274s (1/2)**
>
> We thank the reviewer for the valuable and constructive comments, and address them in detail below.
>
> >W1 & W2: Lack of the new insight and technical contribution of Video-3DGS.
>
> The primary goal of Video-3DGS is to enhance temporal consistency and smoothness in video synthesis by reconstructing video. To the best of our knowledge, this is the **first work** to leverage a 3D Gaussian representation for monocular video reconstruction and then exploit this representation for monocular video editing. Our approach is both fundamental and technically sound, as summarized below.
>
> 1. **Addressing the Core Limitations of 3DGS in Monocular Video Scenes**: Unlike prior methods [1] that rely on external depth estimators (replace COLMAP’s 3D point initialization with depth estimation), our framework fundamentally tackles the intrinsic limitations of both COLMAP and existing 3D Gaussian representations in dynamic, monocular scenes. Specifically, COLMAP tends to fail when there is background clutter or dynamic foreground objects, and a single set of 3D Gaussian struggles to capture complex scene dynamics (as shown in Table 3 of main paper).
>
> 2. **Non-Trivial Extension of COLMAP (MC-COLMAP)**: To address these challenges, we propose a revised version of COLMAP called MC-COLMAP. It aims to minimize motion and complex background through two key technical solutions: spatial decomposition and temporal decomposition. We emphasize that MC-COLMAP’s decomposition strategy is not merely an engineering effort, but a research-driven approach aimed at resolving the fundamental limitations of 3DGS core components when deployed in dynamic scenes.
>
> 3. **Multiple 3D Gaussian Representations for Dynamic Scenes**: Building on MC-COLMAP, we introduce multiple sets of 3D Gaussians—Frg-3DGS and Bkg-3DGS for each clip—optimized separately for foreground and background points. This design brings several technical contributions: (i) *Spherical-shaped random background points* (conditioned on foreground points), (ii) *A merging operation* integrating Frg-3DGS and Bkg-3DGS (see Eqn.5  in the main paper), (iii) *Clip-level optimization with overlapping frames*. Collectively, these innovations provide robust reconstruction capabilities in dynamic scenes and yield improved performance in video reconstruction benchmarks.
>
> 4. **Exploitation of Reconstruction Ability for Video Editing**: Bridging between video reconstruction and editing requires a careful approach. In that vein, we introduce two specific technical contributions to capitalize on the reconstruction capabilities of Video-3DGS for editing: (i) *Style Adherence While Preserving Structure*: as described by Eqn.6  in the main paper, we propose a novel loss function that adaptively retrains the spherical harmonics and opacity of the pretrained 3DGS. This ensures that the edited videos maintain structural fidelity while adhering to the desired editing style. (ii) *Recursive and Ensemble Refiner*: we further introduce a recursive and ensemble-based refiner that effectively leverages our reconstruction framework within the editing pipeline. This refiner boosts editing quality by repeatedly consolidating and refining reconstruction cues, ultimately delivering more temporally consistent editing results with high fidelity to editing prompt.
>
> Video-3DGS not only addresses the core limitations of 3DGS in dynamic monocular video but also advances a new refining paradigm for editing by reconstructing video—one that goes beyond mere system integration. We believe these contributions collectively demonstrate the originality and impact of our work.
>
> [1] COLMAP-Free 3D Gaussian Splatting, Yang et al.

---

> ### Author Response · Authors · 2025-01-10
> **To Reviewer 274s (2/2)**
>
> > Requested Change: additional comparison in video reconstruction and editing
>
> Thank you for suggesting additional experiments. We address them below.
>
> **Video Reconstruction with Other NeRF and 3DGS Methods**
>
> * Robust-Dyn [1]: As reported in Table 1 of main paper, we have already included the video reconstruction results of Robust-Dyn on the DAVIS benchmark. Our Video-3DGS not only achieves higher reconstruction accuracy than Robust-Dyn but also runs significantly faster.
>
> * Shape-of-Motion [2]: We did not include Shape-of-Motion because its rendering quality is significantly lower than the qualitative results reported in the original paper (quantitative result of DAVIS dataset is not reported in paper). Reproduction challenges are also publicly issued [here](https://github.com/vye16/shape-of-motion/issues?q=davis). In particular, our reproduced results show that Shape-of-Motion achieves only **17.9 PSNR** on DAVIS, compared to our Video-3DGS's 3k iteration result (**37.6 PSNR**). Moreover, Shape-of-Motion requires five preprocessing steps (mask estimation, metric depth, monocular depth, camera estimation, and 2D tracking), whereas our approach only needs foreground extraction.
>
>
> **Video Editing with Other Video Editors**
>
> * Instruct-nerf2nerf [3] and Gaussian Editor [4]: These two methods are not inherently designed to edit monocular video. Instead, we have compared against Text2Video-Zero, which leverages the Instruct-Pix2Pix editing pipeline. We demonstrate that our Video-3DGS can serve effectively as a refiner with instruct-pix2pix as shown in Table 2 and Figure 6 of main paper, already providing an approximate comparison to Instruct-nerf2nerf.
> * Evaluation with another video editor: To address the reviewer’s request for additional baselines, we also compared against the state-of-the-art zero-shot video editor, Slicedit [5]. Our Video-3DGS improves the temporal consistency of Slicedit from **0.924 WarpSSIM** to **0.945 WarpSSIM** on DAVIS dataset, while maintaining comparable text fidelity.
>
> In summary, our Video-3DGS demonstrates its effectiveness and practicality by outperforming several existing methods. In the video reconstruction task, we compared against seven different reconstruction methods, and in the editing task, we showed that Video-3DGS functions effectively as a plug-and-play refiner alongside five different video editors.
>
> [1] Robust Dynamic Radiance Fields, Liu et al.
>
> [2] Shape of Motion: 4D Reconstruction from a Single Video, Wang et al.
>
> [3] Instruct-NeRF2NeRF: Editing 3D Scenes with Instructions, Ayaan et al.
>
> [4] GaussianEditor: Editing 3D Gaussians Delicately with Text Instructions, Fang et al.
>
> [5] Slicedit: Zero-Shot Video Editing With Text-to-Image Diffusion Models Using  Spatio-Temporal Slices, Cohen et al.

---

> ### Author Response · Authors · 2025-02-14
> **Summary of the Changes in Revised Manuscript**
>
> We sincerely appreciate the valuable feedback from Reviewer 274s. We have carefully considered all the suggestions and made the necessary revisions to our manuscript. The following is a summary of a key modification and you can check our rebuttal to your requested change reflected in revised version.
>
> > Requested Change: additional comparison with other video editors.
>
> Following the recommendation from the reviewers 274s and k6rY, we conducted additional experiments on recent video editors to show strong compatibility of our Video-3DGS as you can refer to **Subsubsection of D.2 "Comparison With Editors Using Video Generation Models" on Page 21 in Appendix**.

---

### Review · Reviewer_9M7g · 2025-01-06

**Summary Of Contributions:**

This paper addresses the challenge of zero-shot video editing by introducing a novel method called Video-3DGS, a 3D Gaussian splatting-based video refinement technique designed to improve temporal consistency in zero-shot video editors. Video-3DGS represents video content as point clouds, separating it into foreground and background components, which are further modeled as two sets of 3D Gaussians. This point cloud decomposition provides reconstruction constraints that guide a pre-trained diffusion model to maintain temporal coherence in edited videos. The proposed approach achieves high reconstruction quality and efficient training, effectively enhancing the temporal consistency of zero-shot video edits.

**Audience:**

Yes

**Claims And Evidence:**

Yes

**Requested Changes:**

**Questions**
- Specifically, what is the condition for the MC-COLMAP to return a successful / failed status?
- Are there some scenarios where the video decomposition might fail (despite the MC-COLMAP)? If so, could you provide examples of failed video decompositions to better understand the extent of their impact?
- How many frames can the method edit? How does the number of frames impact the quality of the editing?
- The explanation of the second stage for video editing in Section 2.3 is somewhat challenging to follow. I recommend rewriting it to provide a clearer and more intuitive description of the method.

**Strengths And Weaknesses:**

**Strengths**
- The approach of decomposing the video into point clouds appears original and effective for content reconstruction.
- The reconstruction quality surpasses that of chosen baselines while requiring less training time.
- Comprehensive evaluation and detailed ablation studies.

**Weaknesses**
- The proposed method requires an off the shelf panoptic segmenter to separate background and foreground.
- The quality of edited videos is constrained by the success of the quality of the content decomposition into point clouds.
- Due to the reconstruction objective to build the point clouds, the method doesn’t allow for shape modification of edited objects.

---

> ### Author Response · Authors · 2025-01-10
> **To Reviewer 9M7g (1/2)**
>
> We deeply appreciate your insightful feedback, and respond to each of your comments as below.
>
> >W1: Dependency on off-the-shelf panoptic segmenter.
>
> We acknowledge that our Video-3DGS requires a preprocessing step for foreground object extraction using an off-the-shelf panoptic segmenter. However, this is notably simpler compared to the five-stage preprocessing pipeline of shape-of-motion [1], which involves mask estimation, metric depth, monocular depth, camera estimation, and 2D tracking. Furthermore, we studied the influence of different variants and settings of the off-the-shelf segmenter on video reconstruction quality to ensure robustness. There are several open-vocabulary video object segmentation models available, and we have chosen the most representative models for our experiments, including the DEVA model with SAM, and Mobile-SAM. We experimented with three settings: (a) DEVA with SAM and user prompts (i.e., user needs to provide the object class names), (b) DEVA with SAM and VIPSeg foreground categories (i.e., we use the foreground categories defined by VIPSeg dataset without any user prompts), and (c) DEVA with Mobile-SAM and VIPSeg foreground categories. As shown in this [video](https://anonymous-video-3dgs-seg-ablation.github.io), our Video-3DGS is robust to the change of segmentation modules.
>
> >W2: Video editing proposed by Video-3DGS is constrained by the success of MC-COLMAP.
>
> We appreciate the reviewer's concern regarding the dependence of the MC-COLMAP clip operation on the success of reconstruction and editing of video. Extending COLMAP for use with 3D Gaussians in dynamic scenes is indeed a challenging task. However, we have made significant efforts to quantify and improve its performance, as demonstrated by the extensive empirical results presented in our paper. While the success of MC-COLMAP is not entirely controllable, our modifications aim to enhance its robustness in dynamic scenarios. We acknowledge that there are limitations and potential failure cases, but our empirical results show a considerable improvement in success rates, underscoring the effectiveness of our approach. We are committed to further refining our method to address these challenges and enhance its reliability in future work.
>
> >W3: Due to the reconstruction objective to build the point clouds, the method doesn’t allow for shape modification of edited objects.
>
> Thank you for pointing out the limitation. Actually, we already acknowledged this limitation as written in Appendix F. Unsatisfactory outcomes can occur if substantial alterations to object shapes are required,  which conflict with the core principle of 3D Gaussians that aims to maintain the original structure. Nonetheless, our work marks a pioneering step toward extending 3D Gaussian representations to improve the temporal consistency of initially edited videos, regardless of the zero-shot video editing method used.
>
>
> [1] Shape of Motion: 4D Reconstruction from a Single Video, Wang et al.

---

> ### Author Response · Authors · 2025-01-10
> **To Reviewer 9M7g (2/2)**
>
> > Q1: Specifically, what is the condition for the MC-COLMAP to return a successful / failed status?
>
> MC-COLMAP is designed to return failed status when structure-from-motion (SfM) cannot find consistent point matches across video frames. This issue arises from two main factors:
> 1. **Noisy Background**: Excessive noise in the background can confuse the SfM matcher, leading to point mismatches. We mitigate this issue by eliminating background with off-the-shelf segmenter. Then, we set the pseudo background conditioned with extracted foreground points to be used as input of Bkg-3DGS. We showed the effectiveness of Bkg-3DGs with pseudo background points in Table 3.
> 2. **Extensive Foreground Object Motion**: Foreground objects move dynamically across whole video frames, hindering consistent point matching using single view. To address this, MC-COLMAP divides the video into multiple clips and progressively initializes 3D points for each segment, improving SfM robustness for dynamic objects.
>
> > Q2: Failure Cases of MC-COLMAP.
>
> Below scenarios illustrate when MC-COLMAP can struggle or fail entirely.
>
> 1. **Significant Occlusions of Foreground Objects**: When a foreground object becomes fully or partially occluded in consecutive frames, MC-COLMAP’s decomposition process can struggle to track and re-identify the object. As a result, the reconstructed point cloud may contain significant gaps or artifacts. This situation commonly occurs when another object or the scene geometry itself blocks the foreground entity for a considerable duration of the sequence.
>
> 2. **Excessive Motion Blur**: Rapid camera or object movements can induce motion blur, making it difficult for feature detection algorithms to match points across frames accurately. If the blurred regions prevent consistent feature extraction, the reconstruction pipeline may fail to build a coherent model for certain frames or produce large alignment errors.
>
> > Q3: How many frames can the method edit? How does the number of frames impact the quality of the editing?
>
> In Table 2, we evaluated the effectiveness of our Video-3DGS approach as a video-editing refiner on multiple benchmark datasets, covering scenarios that range from shorter sequences of **32 frames (DAVIS)** to longer ones with up to **128 frames (YouTube)**. Our experiments indicate that Video-3DGS can maintain robust performance across these different sequence lengths, provided the initial video editor is capable of handling the corresponding number of frames. A key advantage of our method lies in its ability to seamlessly stitch edits across different segments of the video. Specifically, overlapping frames between clips are jointly optimized with the neighboring 3DGS regions, ensuring smooth transitions and consistent edits throughout the entire video. This architectural design allows Video-3DGS to scale effectively to longer sequences as well, with minimal risk of compromising editing quality or coherence.
>
> > Q4: The explanation of the second stage for video editing in Section 2.3 is somewhat challenging to follow. I recommend rewriting it to provide a clearer and more intuitive description of the method.
>
> Thank you for pointing out the unclear section. We will address it in our final manuscript by providing further details to ensure clarity and accuracy. We appreciate your valuable feedback.

---

> > ### Author Response · Authors · 2025-02-14
> > **Summary of the Changes in Revised Manuscript**
> >
> > We appreciate Reviewer 9M7g's insightful feedback. After thoroughly reviewing your suggestions, we have updated our manuscript accordingly. Below is a summary of the key changes that aim to reflect your questions.
> >
> > > Q1: Specifically, what is the condition for the MC-COLMAP to return a successful / failed status?
> >
> > Please see **Section of Disscusion & Limitations on Page 23 in Appendix** for more analysis of failure case in MC-COLMAP.
> >
> > > Q4:  The explanation of the second stage for video editing in Section 2.3 is somewhat challenging to follow. I recommend rewriting it to provide a clearer and more intuitive description of the method.
> >
> > We reformatted the **writing style of Section 2.3 and place of Figure 5 on page 7 and 8** for clearer and more intuitive description of the method.

---

### Review · Reviewer_k6rY · 2025-02-04

**Summary Of Contributions:**

The paper introduces Video-3DGS, a novel approach that enhances temporal consistency in video editing by leveraging 3D Gaussian Splatting (3DGS). The method follows a two-stage optimization process: (1) video reconstruction, where it refines dynamic monocular videos using a modified MC-COLMAP pipeline to extract structured foreground and background 3D representations, and (2) video editing, where the refined video representations serve as constraints to improve the temporal stability of zero-shot video editors.

**Audience:**

Yes

**Claims And Evidence:**

No

**Requested Changes:**

Please provide results with more recent video editing baselines, as mentioned above

**Strengths And Weaknesses:**

Strengths:
1. The method effectively reduces flickering and inconsistent object rendering in video editing through structured 3D constraints.
2. The method is a plug-and-play module that is compatible with off-the-shelf video editing frameworks.

Weaknesses:
1. Can the author provide more intuition on why it is necessary to introduce MC-COLMAP for 3D reconstruction and then do Gaussian splatting from 3D back to 2D in video editing refinement? Will the error accumulate during this 2D-3D-2D process?
2. While it makes sense that video editing methods based on zero-shot adaptation of image diffusion models (e.g. Tokenflow, T2V-Zero) lack the video generation prior and can thus introduce temporal artifacts, recently there has been a trend of using video generation models (e.g. text-to-video generation model or image-to-video generation model) as a prior for video editing. Examples include UniEdit [1] (using a T2V model), AnyV2V [2] and I2VEdit [3] (using I2V models). These methods effectively overcome the lack of temporal understanding problems and reduce the flickering artifacts. It is necessary to compare Video-3DGS against these methods and see if the proposed 3D reconstruction and Gaussian splatting pipeline outperform these methods (or if using them together can further enhance the editing quality).

[1] Bai, Jianhong, et al. "Uniedit: A unified tuning-free framework for video motion and appearance editing." arXiv preprint arXiv:2402.13185 (2024).
[2] Ku, Max, et al. "AnyV2V: A Tuning-Free Framework For Any Video-to-Video Editing Tasks." Transactions on Machine Learning Research (2024).
[3] Ouyang, Wenqi, et al. "I2VEdit: First-Frame-Guided Video Editing via Image-to-Video Diffusion Models." SIGGRAPH Asia 2024 Conference Papers. 2024.

---

> ### Author Response · Authors · 2025-02-14
> **To Reviewer k6rY**
>
> We sincerely appreciate your detailed insights and have addressed every one of your points as follows.
>
> > W1:  More intuition on why it is necessary to introduce MC-COLMAP for 3D reconstruction and do Gaussian splatting from 3D back to 2D in video editing refinement.
>
> Video-3DGS aims to enhance the temporal consistency of video editing by leveraging 3D Gaussian representations with a two-stage approach. Specifically, in the 1st stage, our Video-3DGS reconstructs videos using 3D Gaussians to establish temporal correspondence between frames within each clip and link neighboring clips with overlapping frames. To achieve robust 3D Gaussian initialization, we propose MC-COLMAP, a revised version of COLMAP, specifically designed to be aware of foreground and background and progressively process video frames. Table 1 in main paper motivated us to prove the video reconstruction capability of Video-3DGS, showcasing its ability to preserve the content and structure of original raw video without need of any 3D task exploration. Please refer to Section **“Importance of 3D Gaussians for Video Editing Task”** in Section 3.2 of our paper, as well as the linked videos (https://anonymous-video-3dgs-time-correspond.github.io and https://anonymous-video-3dgs-tracking.github.io), where we demonstrate that Video-3DGS exhibits strong temporal correspondence across video frames. This robust temporal consistency of 2D videos from Video-3DGS motivates the second stage of our approach, where we can effectively utilize the capability of temporal correspondence for video editing refinement. During this stage, we optimize only the spherical coefficients (SH) while keeping the positional and deformation parameters fixed, ensuring that the 3D-to-2D projection maintains structural integrity without introducing additional errors.
>
> >W2: More comparisons with other methods using video generation models.
>
> We thank the reviewer for recommending other methods that utilize video generation model as prior for video editing. We selected a representative method from each category: UniEdit (using T2V model) and AnyV2V (using I2V model). For implementation detail, since UniEdit requires a huge amount of GPUT memory, we need to split DAVIS videos with 32 frames into 4 chunks. Then, we utilized LaVie [1] T2V model as a prior for video editing. For AnyV2V, we edited the first frame using Instruct-pix2pix (following their default setting) and propagate into other frames in video. After we obtained the initial outputs from each method, we applied our Video-3DGS for refinement. Below are the detailed experiments using DAVIS 16 videos with style-change prompts from LOVEU benchmark.
>
> | DAVIS (480p)                  | WarpSSIM   | Qedit |
> |---------------------------------------|--------|------|
> | UniEdit                                     | 0.779 | 19.95 |
> | + Video-3DGS                 | 0.830 (+5.1%) | 21.15 (+1.2%) |
> | AnyV2V               | 0.737 | 19.85 |
> | + Video-3DGS                          | 0.836 (+9.9%) | 22.25 (+2.4%) |
>
> The results demonstrate that our Video-3DGS also shows good compatibility with recent methods using video generation prior.
>
> [1] LaVie: High-Quality Video Generation with Cascaded Latent Diffusion Models, Wang et al., IJCV 2024.

---

> > ### Author Response · Authors · 2025-02-14
> > **Summary of the Changes in Revised Manuscript**
> >
> > We thank again Reviewer k6rY for the constructive comments. We have carefully considered all of your comments/suggestions and revised our manuscript accordingly. The following summary outlines the key modifications. Please refer to the updated draft and the previous rebuttal for the full discussions on all points.
> >
> > > W2: More comparisons with other methods using video generation models.
> >
> > Please see **Subsubsection of D.2 "Comparison With Editors Using Video Generation Models" on Page 21 in Appendix** for more comparisons with other methods using video generation models.

---

### Decision · Action_Editor_pJCx · 2025-03-17

**Recommendation:** Accept with minor revision

**Comment:**

Scope of Contributions.
- Two reviewers (k6rY and 9M7g) acknowledge that the proposed framework is more engineering-focused yet find its 3DGS-based pipeline for monocular video reconstruction and editing to be valuable. Both highlight the empirical improvements on flicker and temporal consistency, which are long-standing challenges for zero-shot video editing. While reviewer 274s is less convinced of the paper’s novelty, they do not dispute that the approach is carefully validated and obtains improved results.

Empirical Validation.
- The authors addressed the request for additional comparisons with recent video editors (e.g., UniEdit, AnyV2V, Slicedit). The results indicate that Video-3DGS effectively improves temporal stability when combined with these methods. Furthermore, the authors performed additional experiments to benchmark the approach against dynamic reconstruction methods (e.g., Robust-Dyn, partial comparisons with Shape-of-Motion) and showed that 3DGS-based pipeline typically yields higher PSNR in video reconstruction while being faster or requiring fewer complex preprocessing steps.

Addressing Reviewer Concerns.
- Novelty. While 274s still considers the paper’s technical innovations incremental, TMLR guidelines allow acceptance when a system’s thorough empirical validation and demonstration of practical utility make it a worthwhile reference for the community.
- Clarity and Additional Evaluations. Reviewers requested (and the authors provided) clarifications on the second-stage editing process, success/failure conditions of the MC-COLMAP pipeline, and comparisons with more specialized video-editing models. This bolstered confidence in the robustness and general usability of Video-3DGS.

Remaining Minor Points for Revision.
- The text could further emphasize limitations (e.g., shape manipulation is not straightforward with 3DGS, dependence on panoptic segmentation, potential failures in highly dynamic scenes).
- Further clarifications around MC-COLMAP might help readers who wish to replicate the decomposition and understand failure cases in even more detail.

Conclusion.
- With two reviewers leaning toward acceptance and one leaning toward rejection primarily due to novelty concerns, the consensus is that the method and its evidence-based improvements will still interest the TMLR community. The paper should be accepted, pending minor clarifications and polishing in the final draft.

**Audience:**

Two reviewers explicitly state that TMLR’s audience would benefit from learning about the technique. They highlight that bridging 3D-aware methods and 2D diffusion-based editing is both nontrivial and relevant to the video-editing and machine-learning communities. While the third reviewer (274s) questions whether the paper might be more suitable for an applied computer vision venue (citing a lack of fundamental ML novelty), they do not argue that there is no audience interest—only that the work may be more “engineering-oriented.” Given TMLR’s stated criteria (which allow acceptance of work that is sufficiently well-executed and validated, without requiring a groundbreaking technical innovation), there should still be substantial interest in these findings within the TMLR readership.

**Claims And Evidence:**

All three reviewers acknowledge that the paper’s main claim—namely, improving temporal consistency in zero-shot video editing by leveraging a 3D Gaussian splatting (3DGS) pipeline—is supported by the experiments. Two reviewers (k6rY and 9M7g) confirm that the proposed approach demonstrates clear empirical benefits (e.g., mitigating flickering artifacts) across multiple baselines, and the authors have addressed requests for additional comparisons (e.g., with UniEdit, AnyV2V). Although one reviewer (274s) questions the novelty of the method, they do not dispute that the stated objectives (enhancing temporal consistency via 3D reconstruction) are met. Overall, the paper includes sufficient experiments (quantitative and qualitative) that convincingly support its claims of improving temporal consistency.